

# Cloud vertical distribution from combined surface and space radar/lidar observations at two Arctic atmospheric observations

Yinghui Liu[1], Matthew D. Shupe[2], Zhien Wang[3], and Gerald Mace[4]

[1]Cooperative Institute of Meteorological Satellite Studies, University of Wisconsin at Madison, Madison, WI, USA
[2]Cooperative Institute for Research in Environmental Sciences, University of Colorado and NOAA Earth System Research Laboratory, Boulder, CO
[3]University of Wyoming, Laramie, Wyoming
[4]University of Utah, Salt Lake City, UT

*Correspondence to*: Yinghui Liu (yinghuil@ssec.wisc.edu)

**Abstract.** Detailed and accurate vertical distributions of cloud properties (such as cloud fraction, cloud phase, and cloud water content) and their changes are essential to accurately calculate the surface radiative flux and to depict the mean climate state. Surface- and space-based active sensors including radar and lidar are ideal to provide this information due to their superior capability to detect clouds and retrieve cloud microphysical properties. In this study, we compared the annual cycles
of cloud property vertical distributions from satellite active sensors and surface-based active sensors at two Arctic atmospheric observation stations, Barrow and Eureka. We used this data to identify the sensors' respective strengths and limitations and to develop a blended cloud property vertical distribution by combining both sets of observations. Results show that surface-based observations offer a more detailed cloud property vertical distribution from the surface up to 11 km above mean sea level (AMSL) with limitations in the middle and high altitudes; the annual mean total cloud fraction from
space-based observations see 25-40% fewer clouds below 0.5 km than that from surface-based observations, and space-based observations also show much less ice cloud and mixed phase cloud, and slightly greater liquid cloud from the surface to 1 km; space-based observations show comparable cloud fraction between 1 km and 2 km AMSL, and greater cloud fraction above 2 km AMSL than that from surface-based observations. The blended product combines the strength of both products to provide a more reliable annual cycle of cloud property vertical distribution annual cycle from the surface to 11
km AMSL. This information can be valuable for deriving an accurate surface radiative budget in the Arctic and for cloud parameterization evaluation in weather and climate models.

## 1 Introduction

The Arctic has changed dramatically in recent decades, and the causes of these changes and their feedbacks to the global climate system are under intense investigation. The Arctic is warming at a higher rate than that of the global average, a
phenomenon known as Arctic amplification (Solomon et al. 2007, Serreze and Francis 2006); Arctic sea ice extent has been decreasing dramatically (Serreze et al. 2015), a trend that is expected to continue (Holland and Bitz 2003, Overland and


Wang 2013). Changes in the Arctic have likely led to changes in the weather and climate in the midlatitudes through teleconnections in the large-scale circulation (Francis and Vavrus 2012). By studying the factors influencing the Arctic climate system and its changes, we will improve understanding of the Arctic climate and its relationship to the global climate system.

The largest uncertainty in predicting the Arctic climate arises from our lack of understanding of the role clouds play in the Arctic climate system (Solomon et al. 2007, Boucher et al. 2013). A complete, accurate description of three dimensional cloud properties is critical to determine the radiation flux both at the surface and at the top of atmosphere (TOA), as well as the radiative heating rate in the atmosphere. Examining and understanding changes in these vertical distributions are key to studying the recent Arctic changes.

Cloud products from space-based radar/lidar combined observations have the potential to provide comprehensive information on the vertical distribution of cloud properties. These observations have been used to describe global cloud spatial distributions and their temporal changes (Li et al. 2015, Naud et al. 2015). However, space-based low cloud observations are limited by radar ground clutter and strong attenuation of lidar signals, especially by liquid and mixed phase clouds (Marchand et al. 2008; Blanchard et al. 2014). Radar reflectivity from CloudSat has been used to generate high

vertical resolution longwave and shortwave radiative flux profiles and corresponding heating rates (L'Ecuyer et al. 2008); assessing the product's accuracy shows that CloudSat's weakness in detecting low clouds introduces the largest uncertainty. This product has been improved by the inclusion of complementary cloud and aerosol information mainly from space-based lidar observations (Henderson et al. 2013). Complementing the space-based observations, surface observations have superior performance near the surface (Shupe et al. 2011, Shupe 2011, Zhao and Wang 2010) and in resolving the diurnal

cycle, with a relatively weaker performance in the middle and upper levels and with poor spatial coverage.

Efforts have been made to investigate the differences in cloud fraction/frequency from surface-based and space-based radar-lidar combined observations and their impact on the radiative fluxes at multiple surface stations. Using such observations, Protat et al. (2014) studied the cloud occurrence frequency around Darwin, Australia and found that space-based observations underestimated the cloud occurrence frequency below 2 km above mean sea level (AMSL) (all heights in the

text hereinafter are in km AMSL), while surface observations do not detect most of the cirrus clouds above 10 km. Blanchard et al. (2014) investigated the difference in cloud fraction and vertical distribution at Eureka, Canada in the Arctic from surface and space-based combined radar-lidar observations from 2006 to 2010. Among many valuable findings, they found that space-based radar–lidar measurements can depict a complete picture of the cloud vertical profile down to 2 km. Mioche et al. (2015) compared vertical profiles of cloud occurrences from surface lidar and space-based lidar, radar, and

combined lidar and radar over the Ny-Ålesund station during March and April 2007, and showed similar results above 2 km as those in Blanchard et al. (2014).

This study focuses on further examining and comparing the performance of space-based and surface based radar-lidar observations and retrievals to capture the vertical distribution of cloud properties, including cloud fraction, cloud phase, and cloud water content, at two Arctic atmospheric observation stations, Barrow, Alaska and Eureka, Canada. Since cloud phase



has been shown to have a particularly strong impact on Arctic cloud radiative effects on the surface (Shupe and Intrieri 2004), it is particularly important to understand how differences in viewing geometry impact observations of different cloud phases. Based on the comparison performed here, this study also proposes blended products of cloud property vertical distributions from surface and space-based cloud observations at those two Arctic sites to serve as a best estimate cloud
product for model and reanalysis evaluation.

## 2 Data and Method

From the possible Arctic atmospheric observation sites, we have selected Barrow (71°19' N, 156°37' W) and Eureka (80°80' N, 85°57' W) because of the availability of daily cloud vertical profiles from surface observations from 2006 to 2010 that are coincident with space-based observations.  The combined radar-lidar cloud fraction best estimation, cloud fraction vertical
profiles, cloud phase vertical profiles, and cloud water content vertical profiles, from surface observations at these two sites are described in detail in Shupe et al. (2011), Shupe (2011), and Shupe et al. (2015). These products are based on coincident measurements from the Ka-band cloud radar, depolarization lidars including the micropulse lidar (MPL) at Barrow and the high spectral-resolution lidar (HSRL) at Eureka, microwave radiometer, and radiosondes, which are combined to determine cloud phase (Shupe 2007) and microphysical properties at 1-min temporal and 100-m vertical resolutions.
Observations from *CloudSat* and the Cloud–Aerosol lidar and Infrared Pathfinder Satellite Observation (CALIPSO) provide an unprecedented opportunity for a spatially extensive picture of cloud cover in the Arctic (Stephens et al. 2002; Winker et al. 2003). The Vertical Feature Mask (VFM) from CALIPSO's Cloud-Aerosol LIdar with Orthogonal Polarization (CALIOP) provides cloud vertical distribution in up to 10 layers at 5 km and 1 km resolutions, and up to 5 layers at 1/3 km resolution (Vaughan *et al* 2009). Compared to the 1 km resolution data, the 5 km resolution product can identify weaker
cloud features using an iterative multi-resolution averaging scheme (Vaughan et al. 2009). The CALIPSO products from June 2006 to December 2010 were obtained from the Atmospheric Science Data Center at NASA Langley Research Center. The Cloud Profiling Radar (CPR) onboard CloudSat also provides echo mask, in variable "CPR_Cloud_mask" at 125 vertical range bins in a product known as the Level 2 geometrical profiling product (2B-GEOPROF) (Marchand et al. 2008). The latest CloudSat cloud mask (R04) has negligible surface about 0.96 km above the surface. Due to the surface clutter,
only strong cloud or precipitation signals can be detected in the lowest approximately 0.7 km, while weaker cloud signals are missed. In this study, a range bins is defined as cloud with CPR_Cloud_mask equal to and larger than 20, which include weak echo, good echo, and strong echo. Very weak echo, and echo with likely surface clutter are not included. The Radar–Lidar Geometrical Profile Product (2B-GEOPROF-lidar; Mace et al., 2014) merges the CloudSat GEOPROF (Marchand et al., 2008) and the CALIPSO VFM (Vaughan et al., 2009). 2B-GEOPROF-LIDAR contains parameters for up to five
hydrometeor layers, including the cloud base and top heights above mean sea level of each hydrometeor layer in one radar footprint along with the longitude and latitude.





A level 2 combined product, 2B-CLDCLASS-lidar, combines CPR and CALIOP measurements for cloud phase determination into eight basic cloud types (Sassen and Wany 2012). Ice, water/liquid, and mixed phase clouds are identified for up to 10 layers. 2B-CLDCLASS-lidar collocates CALIOP L1 measurements to CPR footprints, then determines cloud vertical structures (Wang et al. 2008) and cloud phase. The microphysical property differences between water and ice

particles, including size, location, falling speed and number concentrations, result in large differences in their radiative properties, and in turn a large difference in the CALIPSO lidar and CloudSat CPR signal. Cloud phase is effectively determined using the different sensitivities of CloudSat radar and CALIPSO lidar to ice crystals and water droplets, together with the cloud top and base temperatures.

Based on the measured CPR radar reflectivity factor, another level 2 product, the CloudSat Radar-Only Cloud Water Content

Product (2B-CWC-RO), estimates cloud liquid and ice water content, and effective radius. Effective radius, and water content are retrieved with the assumption that the radar profile is due to a single phase of water, either liquid or ice. Using a simple scheme based on an ECMWF model temperature profile, this product combines separate liquid and ice profiles into a mixture of ice and liquid phases over the portion of the vertical profile within the proper temperature range. The temperature profile is obtained from ECMWF reanalysis data that have been collocated in space and time to the CloudSat radar profile

and interpolated to the CloudSat vertical resolution. It should be noted that the retrieval is not designed to determine mixed-phase cloud properties directly.

In this study, vertical profiles of cloud fraction from CALIPSO at 1/3 km, 1 km and 5 km horizontal resolution, 2B-GEOPROF, and 2B-GEOPROF-LIDAR, and vertical profiles of cloud phase (ice, liquid, and mixed phase) from 2B-CLDCLASS-lidar, and vertical profiles of cloud effective radius and water content from 2B-CWC-RO are calculated and

examined. Vertical profiles of all these products within 50 km of the two Arctic atmospheric observation sites, Barrow and Eureka, are extracted and archived. The cloud fraction vertical distribution is calculated as follows. For each vertical profile, the entire profile is remapped to 30 m resolution from the surface to 12 km, such that a single cloud layer in a profile from the surface to 0.6 km would flag the vertical grid number from 1 to 20 as cloud covered with other grids as clear. The mean cloud fraction at each vertical grid point is calculated as the ratio of cloud case numbers to the total vertical profile case

numbers at that vertical height in a selected time period. Combination of the cloud layer products at 5 km and 1/3 km provides a complete vertical and horizontal distribution of clouds from CALIPSO (Vaughan et al. 2009, Vaughan et al. 2005). The vertical profiles of cloud fraction from CALIPSO at 1/3 km and 5 km are combined, and shown in section 3. The combined product is referred as CALIPSO 5 km. To compare, the vertical profiles of cloud fraction from CALIPSO at 1/3 km and 1 km are also combined, and shown in section 3. The combined product is referred as CALIPSO 1 km. For cloud

microphysical property vertical distribution, only the portion with valid profile retrievals is remapped to 30 m resolution from the surface to 12 km, such that a single cloud layer in a profile from the surface to 0.6 km would map the cloud phase (ice, water/liquid, or mixed phase) to a grid number from 1 to 20. The mean cloud phase frequency at each vertical grid point is calculated as the ratio of case numbers of each phase to the total case numbers. Mean cloud water content for ice and liquid phases at each vertical grid point is calculated as the mean values of water content from all available retrievals at that





grid when clouds are present. For deriving these statistics, ice in any type of cloud (ice, and mixed phase) is included, while liquid in any type of cloud (liquid and mixed phase) is included.

Surface-based radar, lidar, and radar-lidar combined products are available from June 2006 to December 2010. Details of the collection and process of the data can be found in Shupe (2011), and Shupe et al. (2011, 2015). Surface observations of good

quality are available at Eureka for most of this time period and at Barrow from middle February 2008 to December 2010. To be consistent with the surface data, the space-based results are considered over the same time periods as surface observations available at each site. The monthly mean sample number of the active sensors is a function of latitude in the Arctic, with the fewest at 60° N, gradually increasing to a maximum around 80° N (Liu 2015). Both factors are reflected in the large number of samples at Eureka, with over 6000 total sample numbers in a month at Eureka, and around 1500 total samples at

Barrow in a month.

## 3 Results

### 3.1 Cloud fraction vertical distribution

#### 3.1.1 Barrow

Cloud fraction vertical distributions from surface observations at Barrow (Figure 1a) reveal that cloud fractions are greater

than 30% at each layer below 0.5 km throughout the year except in March and June. In the low level (surface to 2 km), the cloud fraction vertical distributions show maximum values of between 55% and 85% in October and November. In the middle level (2 km to 6 km), most of the cloud fractions are less than 30%, except local maxima greater than 30% in April and November. Minimal cloud fractions of less than 15% occur above 4 km in January, June and September. In the high level (6 km to 12 km), most cloud fractions are less than 20%, except those between 6 km and 8 km in April, August, and

October.

The space-based observations show similar patterns but different values from those of surface observations at Barrow (Figure 2a, 2b, 2c). CloudSat2B-GEOPROF (Figure 2b) shows little cloud below 0.5 km because of the surface clutter issue, limited cloud distribution between 0.5 km and 1 km, and similar patterns as surface observations above 1 km. CALIPSO 5 km (Figure 2a) shows considerably higher cloud fractions than CALIPSO 1 km (Figure not shown) throughout, and both

products show some cloud fraction distribution below 0.5 km. 2B-GEOPROF-LIDAR (Figure 2c) cloud vertical distribution merges information from both CloudSat and CALIPSO, thus providing a more complete vertical distribution than either of those two alone. Based on the 2B-GEOPROF-LIDAR cloud vertical distribution, the cloud fraction below 0.5 km is less than 30% most of the year, except in May, and November when the local maximum is greater than 30%. In the low level, cloud fraction increases with height, reaches a maximum between 1 km and 1.5 km, and starts to decrease in general. The

annual minimum cloud fraction at this level of less than 20% appears in June and July. In the middle level, cloud fractions are mostly between 20% and 40%. The maximum cloud fraction appears in April, August, and December with values greater





than 35%. The minimum appears in March and June with values less than 16%. In the high level, cloud fraction of 20% or greater appears most of the time except for November, March and June.

Comparison of cloud vertical distributions from space-based observations and surface observations at Barrow show overall least cloud fraction from CALIPSO 1 km, then CALIPSO 5 km, and 2B-GEOPROF, and overall most cloud fraction from

2B-GEOPROF-LIDAR above 1 km, and all space-based cloud fractions are less than that from surface observations in the lowest 1 km (Figure 2, Figure 3). Compared to cloud fraction vertical distribution from surface observations, CALIPSO 1 km shows less cloud fraction in every month from surface to 6-11 km depending on month (Figure not shown); CALIPSO 5 km shows less cloud fraction from surface to 5 km in every month, and greater cloud fraction above 6 km in most months; Above 1 km 2B-GEOPROF has differences from the surface observations of +20 to -10%. In most months 2B-GEOPROF-

LIDAR tends to have greater cloud fractions above 1 km; all space-based cloud fractions show lower cloud fractions below 1 km, with the least from 2B-GEOPROF, then CALIPSO 1 km, CALIPSO 5 km and 2B-GEOPROF-LIDAR. The near surface cloud distributions from 2B-GEOPROF-LIDAR originate from CALIPSO observations and also show much less cloud fraction distributions below 0.5 km, with differences as high as -67% in October. The difference becomes smaller between 0.6 km and 1.2 km. Above 1.2 km, 2B-GEOPROF-LIDAR shows generally greater cloud fractions (up to 27% in

September at 5 km) than those from surface observations.

Comparison of the annual mean cloud vertical distributions from space-based observations and surface observations shows that all space-based observations have lower cloud fractions in the lowest 1 km, while 2B-GEOPROF-LIDAR and CALIPSO 5 km have higher cloud fractions at some heights above 1 km (Figure 4a). More specifically, compared to surface observations, below 0.5 km the space-based observations see 25-40% fewer clouds than are observed from the surface;

between 1 km and 6 km 2B-GEOPROF and 2B-GEOPROF-LIDAR show slightly greater cloud fraction, while CALIPSO 1 km and 5 km show less cloud fraction; above 6 km, CALIPSO 5 km and 2B-GEOPROF-LIDAR show slightly greater cloud fraction, while CALIPSO 1 km and 2B-GEOPROF show less cloud fraction. For 2B-GEOPROF-LIDAR, the greater cloud fractions above 1 km are due to the combined detection capabilities from CALIPSO 5 km and 2B-GEOPROF. The low cloud fraction from space observations below 1 km can be attributed to surface clutter issue from 2B-GEOPROF near the surface,

and the inability of CALIPSO to penetrate optically thick clouds. Surface reported lower cloud fractions above 1 km might be due to the inability of surface lidar penetrate lower optically liquid and mixed-phase clouds, along with the difficulty to detect optically thin clouds composed of small ice particles by surface radar in the middle and upper levels.

Annual cycle of monthly mean total cloud amount at Barrow shows relatively low values from January to March, and relatively high values (75% and higher) from April to December (Figure 5a). Monthly means from space observations and

surface observations share similarities except 2B-GEOPROF shows much lower fractions in all months, e.g. around 30% in June compared to above 75% from surface observations. The 2B-GEOPROF-LIDAR has the most similar annual cycle to surface observations, with lower monthly means from CALIPSO 5 km, followed by CALIPSO 1 km, and with 2B-GEOPROF showing the lowest values and the largest negative differences from May to September. This is in agreement with those presented in Zygmuntowska et al. (2012) considering the CloudSast does not detect the cloud below



approximately 0.5 km. The larger differences from May to September might be attributed to the relatively higher frequency of clouds below 960 m in that time period (Figure 6), which CloudSat does not detect well near the surface.

Shupe (2011) presents cloud phase vertical distributions at Barrow and Eureka based on combined surface lidar, radar, microwave radiometer, and radiosonde observations (Shupe 2007). The main features of cloud vertical distributions of ice

cloud, liquid cloud, and mixed phase cloud at Barrow from 2006 to 2010 (Figure 7) include the following: Ice clouds are prevalent from the surface up to 9-11 km throughout the year except from the surface to 4.5 km in June, July, and August. The maximum ice cloud fractions occur at low levels from October to April, and at middle levels in April, November, and December with a range between 10% and 30%. At high levels, ice cloud fraction between 10% and 20% appears from June to August. Mixed phase clouds generally occur on average 8-20% at low levels, and on average 2-8% at middle levels. The

maximum mixed phase cloud fractions, up to 57%, appear between the surface and 1 km from September to November. Liquid clouds appear between the surface and 0.8 km in the warm season mainly from May to September, with a maximum liquid cloud fraction of greater than 40% in the lowest 0.4 km in August.

Cloud phase vertical distributions at Barrow derived 2B-CLDCLASS-lidar agree in general with the patterns observed above 1 km from surface observations (Figure 7). At Barrow, ice clouds are common throughout the year from 1 km up to 11 km

except from the surface to 4.5 km from June to August, when the ice cloud fraction are mostly less than 7%. Liquid cloud fraction of greater than 10% appears mainly from the surface to 0.8 km in May, August, September, and November. Mixed phase clouds appear between 1 km and 3.5 km throughout the year. A maximum of up to 55% appears at 1 km in October. Another local maximum between 15% and 30% extends from 1 km to 6 km in August, which is not shown in the surface observations. There is little mixed phase cloud distribution below 1 km.

One major difference between the vertical distributions of ice, liquid, and mixed phase clouds from space-based and surface observations is that the space-based observations show much less ice cloud and mixed phase cloud, and slightly greater liquid cloud from the surface to 1 km (Figure 7). Above 1 km, the two perspectives show similar annual average profiles, with the space observations seeing slightly higher mixed-phase cloud fractions from 3-5km, slightly higher liquid cloud fraction from 0.5-3 km, higher ice cloud fraction at 10 km, and lower ice cloud fractions at 2-6 km (Figure not shown),

although month to month variability can be larger (Figure 7).

Annual cycle of monthly mean ice clouds from surface show greater values throughout the year except January (Figure 9a), similar to the mixed phase cloud amount comparison (Figure 9c). Liquid cloud monthly means from 2B-CLDCLASS-lidar show greater values than those from surface observations in all months except January, June, and July (Figure 9b) because space-base measurements have difficulties to detect mixed phase clouds with low ice concentration.

**3.1.2 Eureka**

All cloud distributions at Eureka show different annual cycles from those at Barrow. Cloud vertical distributions from space-based observations at Eureka are relatively smoother than those from space-based observations at Barrow partly due to





greater sample numbers at Eureka. However, general findings about the differences between space-based and surface observations are similar.

Total cloud fraction vertical distribution at Eureka (Figure 1b) from surface observations shows the largest values (up to 55%) between the surface and 0.5 km, except from June to August when low-level values are less than 25% and profile

maximum values are above 1 km. The maximum cloud fraction at low levels at Eureka is considerably lower than that at Barrow. At middle levels, the cloud fractions are mainly 10-30% with a local maximum greater than 30% from September to November. At high levels, most of the cloud fractions are less than 20%.

For the vertical distributions of total cloud fraction from space (Figure 2d, 2e, 2f), CALIPSO 5 km (Figure 2d) and 1 km (Figure not shown) show similar patterns with greater values in CALIPSO 5 km. Both show limited cloud below 0.5 km. A

local maximum between 4 km and 6 km appears from October to February in CALIPSO 5 km. 2B-GEOPROF (Figure 2e) shows little cloud below 1 km, and detailed cloud information above 1 km, with maximum fractions between 1 km and 4 km from September to December. 2B-GEOPROF-LIDAR (Figure 2f) merges information from CALIPSO and CloudSat, and presents a comparable cloud vertical distribution to that from surface observations, except near the surface. At low levels, the 2B-GEOPROF-LIDAR cloud fractions are less than 40%, with maximum between 30% and 40% from September to

November. At middle levels, a local maximum cloud fraction of between 30% and 35% appears between 2 km and 4 km from September to November; a local minimum cloud fraction of less than 15% appears in March. At high levels, cloud fraction is above 20% from July to November between 6 km and 7.5 km.

Though the total cloud fraction vertical distributions and their annual means at Eureka and Barrow are different (Figure 1, Figure 4), comparison of the space-based cloud vertical distributions and their annual means and those from the surface at

Eureka (Figure 3d, 3e, 3f, and Figure 4b) shows qualitatively the same differences as those at Barrow (Figure 3a, 3b, 3c, and Figure 4a). Whether This the differences found in cloud detection capabilities from CALIPSO 1 km, CALIPSO 5 km, 2B-GEOPROF, and 2B-GEOPROF-LIDAR regarding total cloud fraction vertical distributions based on comparisons with surface observations at Barrow and Eureka can be generalized to he whole Arctic might be worth further investigation.

Annual cycle of monthly mean cloud amount at Eureka from surface observations shows relatively low values of between

56% and 67% from February to August, and high values of between 67% and 81% from September to February (Figure 5b). Monthly means from space-based observations show general increasing cloud amount from March to September, and then start to decrease gradually. 2B-GEOPROF-LIDAR shows comparable monthly means as CALIPSO 5 km, and both are greater than those from CALIPSO 1 km and 2B-GEOPROF, with the least typically from 2B-GEOPROF. All space-based monthly means are noticeably smaller from January to March than those from surface observations, and these negative

differences might be due to the relatively higher frequency of clouds below 960 m only. Monthly means from 2B-GEOPROF-LIDAR and CALIPSO 5 km are greater from June to August compared to surface observations, which is possibly due to the higher frequency of clouds above 960 m only, which surface observations might miss (Figure 6b).

For surface observations at Eureka, ice clouds are the prevalent cloud type from the surface to up to 11 km throughout the year except in June, July, and August when there are few ice clouds from the surface to 3 km (Figure 9). The maximum ice





cloud fraction of up to 40% appears at low levels from November to March. At middle levels, ice cloud fractions are mostly between 15% and 25%, with the exception of lower fractions from June to August. At high levels, ice cloud fractions are mostly below 10% except from July to October. Mixed phase clouds are common at low levels except in July and August, and at middle levels from June to September. A maximum mixed phase cloud fraction between 20% and 30% appears

between the surface and 2 km from September to October. Liquid phase clouds are mainly less than 5% throughout the year except in lowest 0.5 km in September and October.

Cloud vertical distributions of ice cloud, liquid cloud, and mixed phase cloud at Eureka from space-based observations show similar pattern above 1 km as those from surface observations (Figure 9). The major differences between the cloud vertical distributions at Eureka (Figure 8d, 8e, 8f, and Figure 9) are similar to those for Barrow (Figure 7, Figure 8a, 8b, and 8c).

Major differences include: much less ice and mixed phase cloud in the lowest 1 km from space-based observations; greater liquid cloud, and mixed phase cloud above 2 km in the vertical distributions and annual mean of vertical distributions (Figure not shown); comparable monthly mean total cloud amount, higher ice cloud monthly means, lower liquid cloud monthly means, and higher mixed phase cloud monthly means from surface observations than space-based observations.

### 3.2 Blended cloud vertical distribution at Barrow and Eureka

While the cloud fraction vertical distributions at Barrow and Eureka show different patterns, the cloud vertical distribution differences between space-based and surface observations are similar for both stations as detailed in Section 3.1. Surface observations show detailed and higher values in the lowest 1 km; space observations provide little cloud information in the lowest 0.5 km, limited information between 0.5 km and 1 km, and comparable or higher values between 1 km and 2 km. In the middle and upper levels, space observations generally show higher values.

Here we generate a blended cloud fraction vertical distribution for total cloud, ice cloud, liquid cloud, and mixed phase cloud from both surface and space-based observations. Below 1 km, the cloud fraction vertical distribution is from surface observations; from 1 km to 2 km, the cloud fraction vertical distribution is calculated as a linear combination of surface observation and space-based observation, with the weights of the surface observations changing from 1.0 to 0.0 linearly as a function of altitude, and the weights of the space-based observations changing from 0.0 to 1.0 linearly as a function of

altitude; above 2 km, the cloud fraction vertical distribution is from space-based observations.

Figure 10 presents the blended total cloud fraction vertical distributions from 2B-GEOPROF-LIDAR and surface observations at Barrow and Eureka from 2006 to 2010. The blended product provides a complete picture of the cloud fraction vertical distribution, by optimally combining surface and space-based observations. There is no apparent discontinuity in the cloud fraction vertical distribution between 1 km and 2 km at Barrow or Eureka. Figure 11 shows cloud

vertical distributions of total cloud, ice cloud, liquid cloud, and mixed phase cloud from 2B-CLDCLASS-lidar and surface observations at Barrow and Eureka respectively from 2006 to 2010. The blended cloud phase vertical distributions from space-based observations show similar patterns as those from surface observations with more complete distributions in the





middle and high levels. The blended product is smoother for Eureka than for Barrow. The transitions between main surface observations below 1km and main space-based observations above 2 km are smooth for all cloud phases.

### 3.3 Cloud water content

The ice water content and liquid water content vertical distributions from 2B-CWC-RO and surface observations at Barrow

are presented in Figure 12. There is limited information below 1 km from space-based observations. Based on the space-based observations, the ice water content is less than 40 mg m$^{-3}$ throughout the year except higher values of up to 100 mg m$^{-3}$ from May to August, and in December from 2 km to 6 km; the liquid water content have high values of between 150 mg m$^{-3}$ and 300 mg m$^{-3}$ from June to August from 1 km to 3.5 km, and in February, September and October between 1 km and 2 km. Surface observations show low ice water content of 20 mg m$^{-3}$ and less above 4 km, and higher values below 4 km, with

maximum values of 60-100 mg m$^{-3}$ from October to February in the lowest 2 km, and in June and July between 1 km and 3 km; liquid water content show high values of 150-250 mg m$^{-3}$ from May to August from the surface to 5 km, and in September and October from surface to 2 km. The similarity between the ice water content from surface and space-based observations includes that both distributions tend to have higher values in June and July, and from December to February, but at different heights. For liquid water content, both surface and space-based observations show high values from June to

August in the lowest 3.5 km, and in September and October below 2 km.

In Eureka, the ice water content from space-based observations is less than 40 mg m$^{-3}$ throughout the year except values of around 60 mg m$^{-3}$ from August to October from 2 km to 5 km, and in April from 2 km to 6 km as shown in Figure 13. The ice water content from surface observations is also below 40 mg m$^{-3}$ throughout the year except values of between 60 and 80 mg m$^{-3}$ from June to October from the surface to 3 km. Liquid water content from both surface and space-based observations

show low values of 75 mg m$^{-3}$ and less from October to April, and high values from June to August below 3 km, with much higher values from space-based observations.

### 4 Conclusions

This study compares the annual cycles of cloud vertical distributions of total cloud, ice cloud, liquid cloud, and mixed phase cloud from combined surface active lidar/radar observations and from multiple space-based active lidar/radar products at two

Arctic atmospheric observation stations, Barrow and Eureka. The primary conclusions are as follows:

- All space-based active radar/lidar cloud observations have limitations in the lowest 1 km AMSL; the surface measurements have superior performance near the surface, and complements the space-based observations. Surface observations show that the highest total cloud fractions of all cloud, ice cloud, liquid cloud, and mixed phase cloud appear between surface and 1 km. All space-based observations show lower total cloud fractions below 1 km, with least

from 2B-GEOPROF, then CALIPSO 1 km, CALIPSO 5 km, and 2B-GEOPROF-LIDAR. The annual mean total cloud fraction from space-based observations see 25-40% fewer clouds below 0.5 km than that from surface-based



observations. Compared to surface-based observations, space-based observations show much less ice cloud and mixed phase cloud, and slightly greater liquid cloud from the surface to 1 km.

- Above 1 km, space-based observations show similar patterns as surface observations, but different magnitudes for total cloud, ice cloud, liquid cloud, and mixed phase cloud. For total cloud fraction, CALIPSO 1 km shows the lowest values, higher values from CALIPSO 5 km especially above 6 km, and followed by highest magnitude from 2B-GEOPROF mainly in the middle level. 2B-GEOPROF-LIDAR, which merges CALIPSO and CloudSat, provides the closest vertical distribution as that from surface observations. The space observations show greater ice cloud fraction above 9 km, greater liquid cloud fraction in general, and greater mixed phase cloud fraction above 1km.

- For the annual cycle of the total cloud fraction, monthly means from space-based observations are generally lower than those from surface observations. Each perspective has its limitations, with the surface observations missing some high-level cloud and the space-based sensors missing some low-level clouds. Both estimates are likely lower than the true cloud fraction, if those missed clouds are not all overlapping with other clouds. Because low clouds are more prevalent at these locations, the surface-based estimate is likely closer to the true total cloud fraction. Annual cycles of monthly mean cloud occurrence by phase show less ice and mixed phase cloud, and greater liquid cloud from space-based observations.

- A blended cloud fraction vertical distribution with a weighted combination of surface and space-based observations can provide a rather complete description of cloud vertical distribution of total cloud, and ice, liquid, and mixed phase clouds from the surface to 11 km.

Existing space-based cloud distributions in the lowest 1 km do not capture all clouds, especially ice and mixed phase clouds. How these missed clouds in the lowest 1 km affects the radiation flux calculations at the surface and at the top of the atmosphere is a topic of future work. The blended cloud property vertical distribution can be used as an input to a Monte Carlo radiative transfer model for a more accurate surface radiation flux calculation at these sites. A blended cloud property vertical distribution can also be used to evaluate cloud parameterizations in both weather and climate models for improvement (Klaus et al. 2016), to study the Arctic atmosphere-sea ice-ocean interactions (Kay et al. 2009, Taylor et al. 2015, Liu et al. 2012a), and in other Arctic cloud studies (Devasthale et al. 2012, Liu et al. 2012b).

Low-level clouds are frequent in the Arctic and important for the surface radiation balance. While space-based cloud observations from active radar/lidar sensors have been critical for improving our understanding of Arctic clouds and their interactions with other climate components in the Arctic, challenges remain in depicting Arctic low-level clouds from space. Surface observations of clouds at existing Arctic atmospheric observatories and a few field campaigns have provided valuable information on Arctic clouds, especially for studying low-level clouds (Tjernström et al. 2014, Uttal et al. 2002). However, such observations are limited in spatial extent and may not represent pan-Arctic cloudiness. Thus it is critical to combine key information from both space-based- and surface cloud measurements to provide the most comprehensive characterization of Arctic clouds possible and to facilitate further understanding of the Arctic climate system.




**Acknowledgements**

Shupe acknowledges support from the US Department of Energy (DOE) Atmospheric System Research Program (DE-SC0011918) and the National Science Foundation (ARC-0632187). The authors thank Norm Wood, and Ralph Kuehn for their valuable comments on improvement of the paper. Ground-based observations from Barrow were obtained from the

DOE Atmospheric Radiation Measurement Program. Ground-based observations at Eureka were obtained from the NOAA Earth System Research Laboratory and the Canadian Network for the Detection of Arctic Change (CANDAC). The CALIPSO products from June 2006 to December 2010 were obtained from the Atmospheric Science Data Center at NASA Langley Research Center. The 2B-GEOPROF, 2B-GEOPROF-LIDAR, 2B-CLDCLASS-lidar, and 2B-CWC-RO products from June 2006 to December 2010 were obtained from the CloudSat Data Processing Center at the Colorado State

University.

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





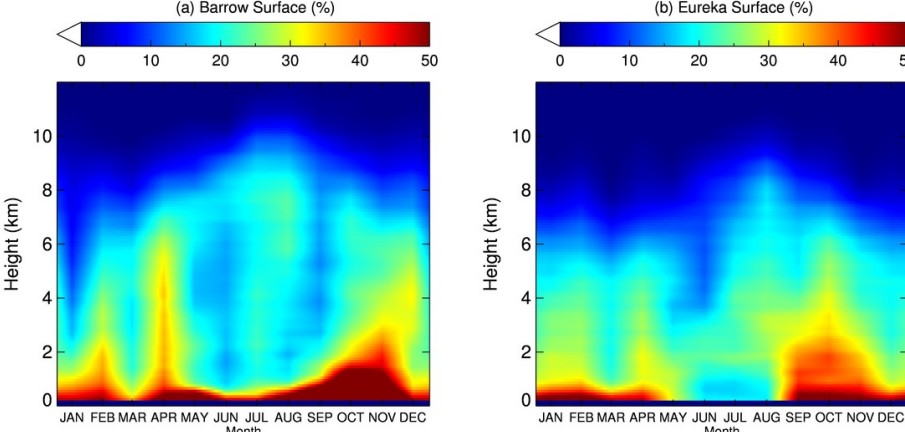

**Figure 1: Cloud fraction vertical distribution from surface observations at (a) Barrow, and (b) Eureka for 2006-2010 (after Shupe et al. 2011).**





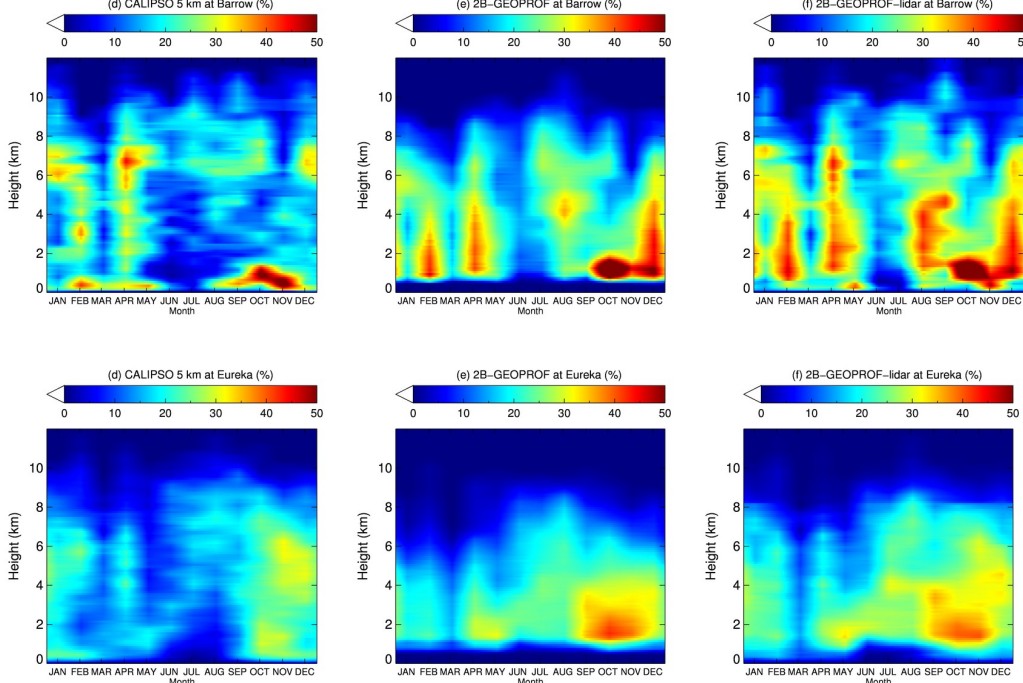

**Figure 2: Cloud fraction vertical distribution 2006-2010 from (a) CALIPSO 5 km, (b) 2B-GEOPROF, and (c) 2B-GEOPROF-lidar at Barrow; (d) CALIPSO 5 km, (e) 2B-GEOPROF, and (f) 2B-GEOPROF-lidar at Eureka.**





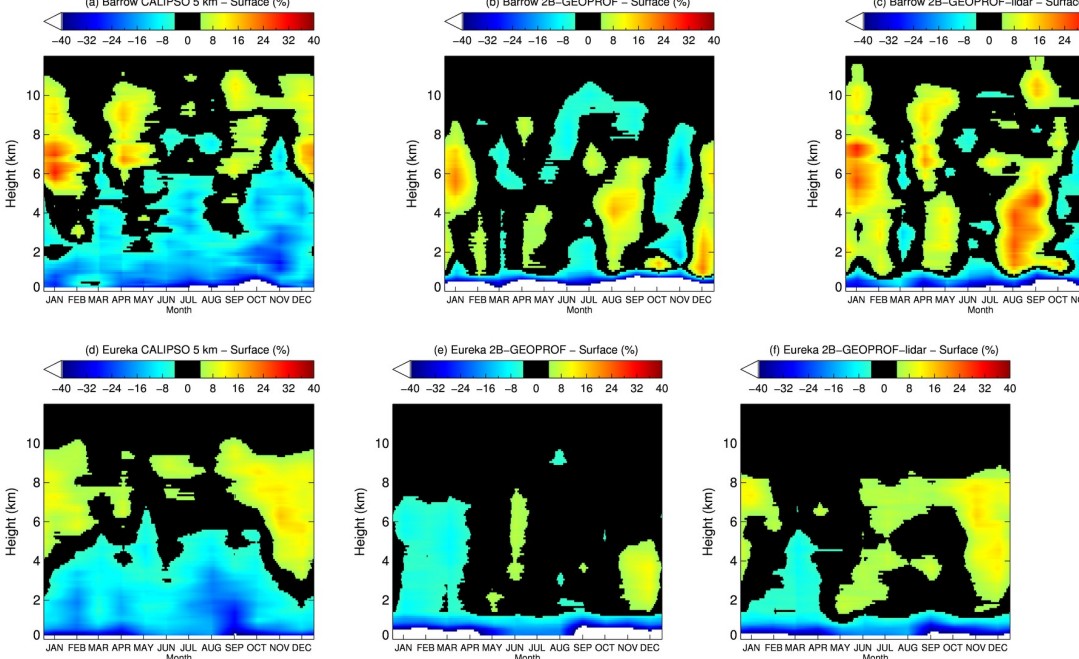

**Figure 3: Cloud fraction vertical distribution difference for 2006-2010 of (a) CALIPSO 5km, (b) 2B-GEOPROF, and (c) 2B-GEOPROF-lidar and surface at Barrow; and of (d) CALIPSO 5 km, (e) 2B-GEOPROF, and (f) 2B-GEOPROF-lidar and surface at Eureka.**





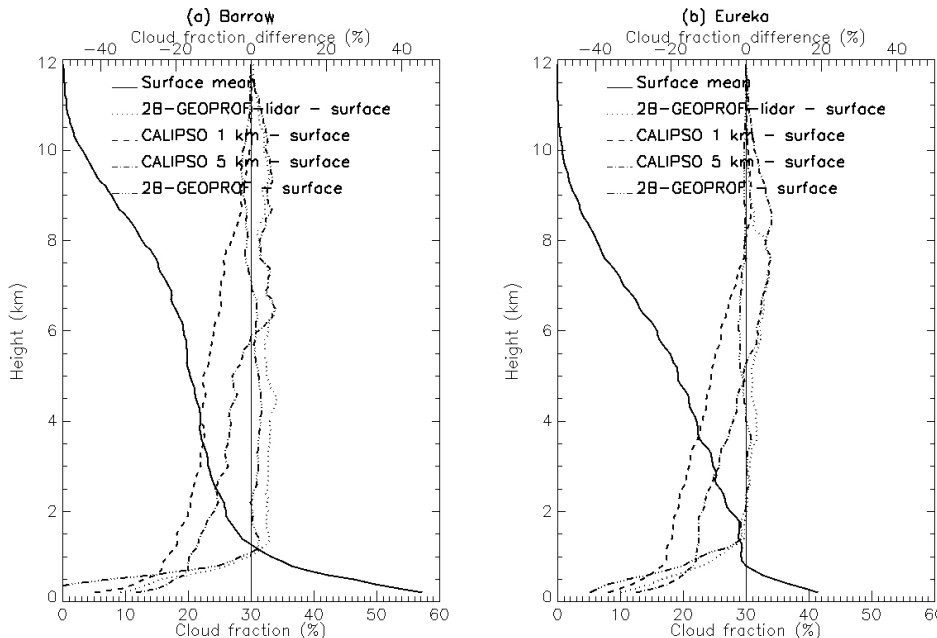

**Figure 4: Mean vertical distributions of cloud fraction from surface, and the difference of 2B-GEOPROF-lidar, CALIPSO 1 km, and CALIPSO 5 km, and 2B-GEOPROF minus surface observations at (a) Barrow, and (b) Eureka for 2006-2010.**





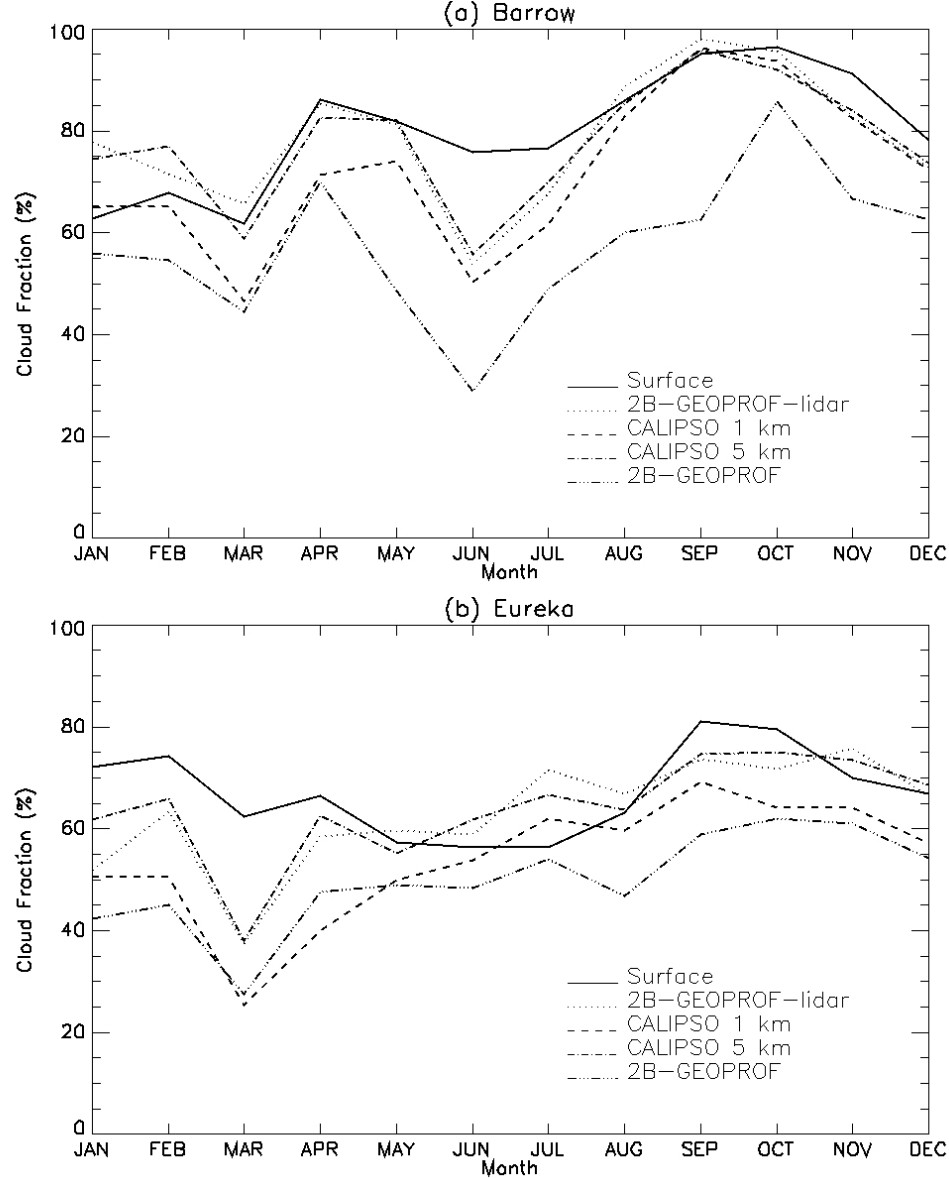

**Figure 5: Monthly mean cloud fraction from surface, 2B-GEOPROF-lidar, CALIPSO 1km, CALIPSO 5 km, and 2B-GEOPROF**
5   **at (a) Barrow, and (b) Eureka (bottom) for 2006-2010.**





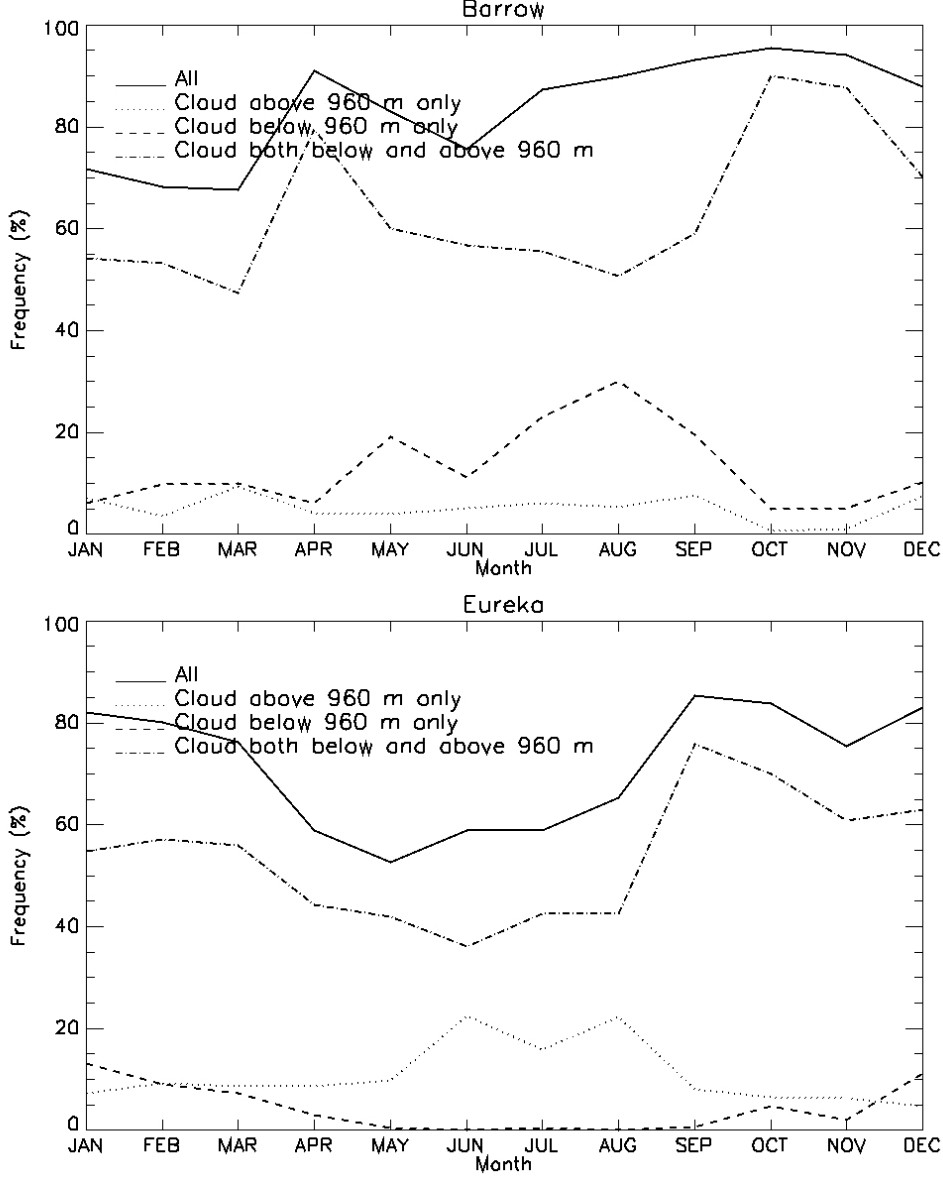

**Figure 6: Mean cloud fraction above 960 m only, cloud below 960 m only, and cloud below and above 960 m from surface observations at Barrow (top) and Eureka (bottom) for 2006-2010.**




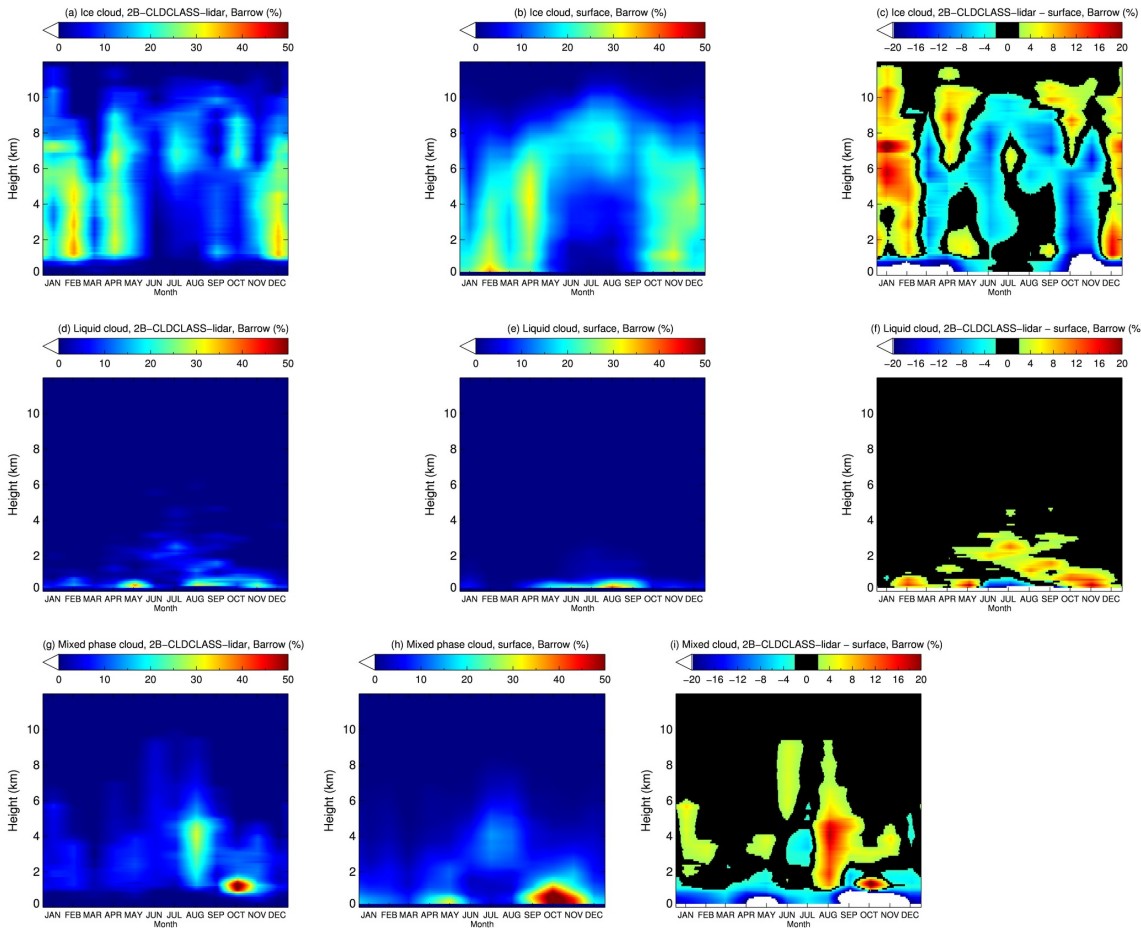

**Figure 7: Vertical Distributions of ice phase cloud (1ˢᵗ row), liquid phase cloud (2ⁿᵈ row), and mixed-phase cloud (3ʳᵈ row) from 2B-CLDCLASS-lidar (left column), from surface (middle column), and difference of 2B-CLDCLASS-lidar and surface at Barrow for 2006-2010.**





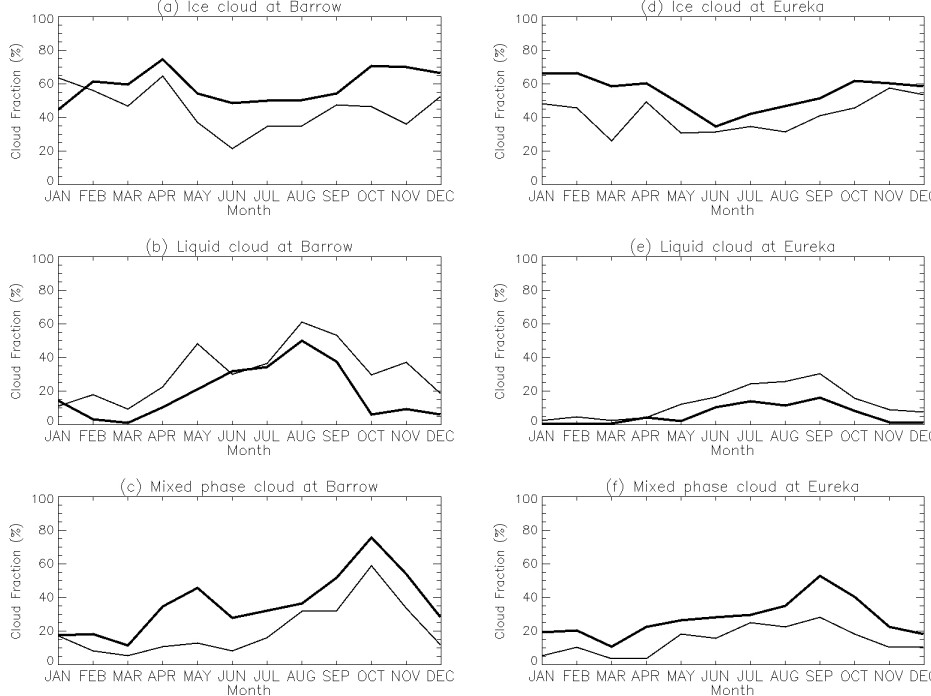

**Figure 8: Monthly mean cloud fraction from surface, and 2B-CLDCLASS-lidar 2006-2010 (a) ice cloud, (b) liquid cloud, and (c) mixed phase cloud at Barrow; 2010 (d) ice cloud, (e) liquid cloud, and (f) mixed phase cloud at Eureka.**





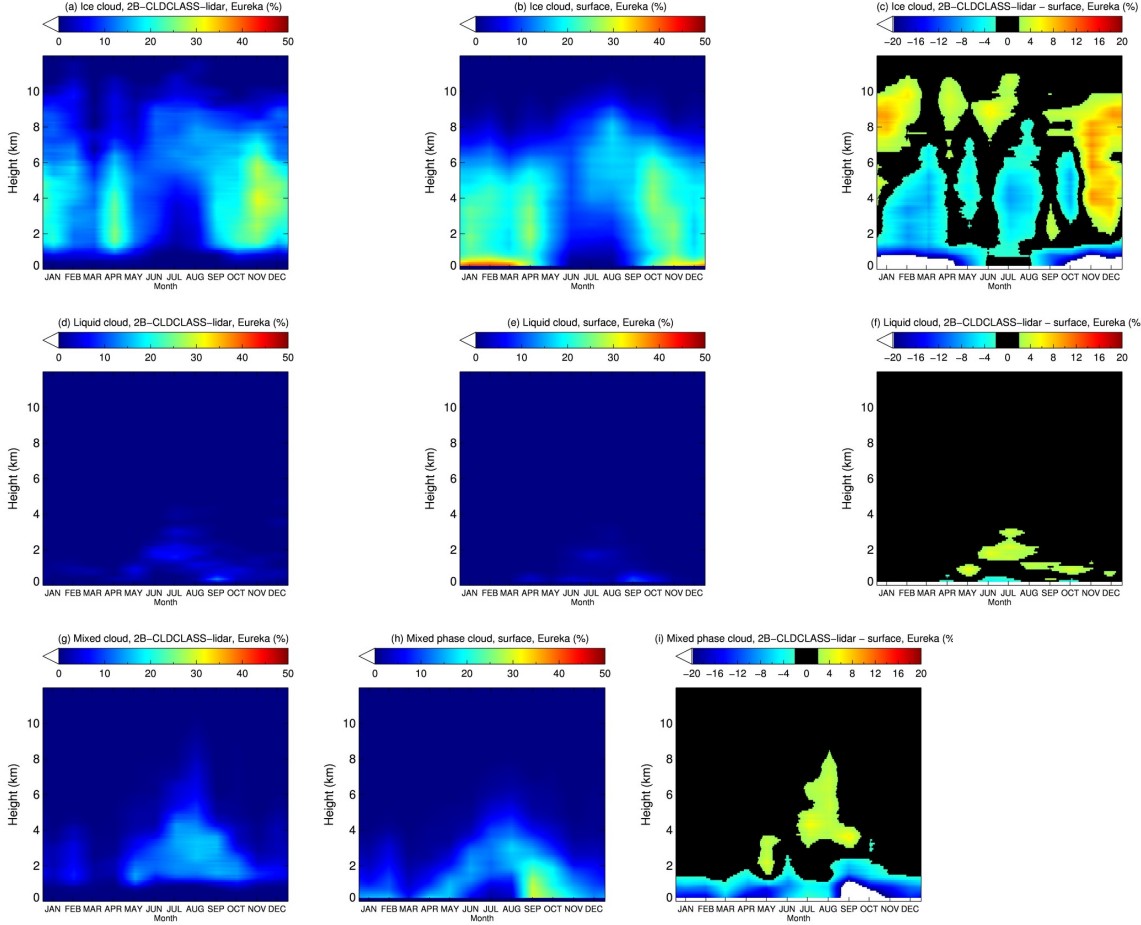

**Figure 9: Vertical Distributions of ice phase cloud (1ˢᵗ row), liquid phase cloud (2ⁿᵈ row), and mixed-phase cloud (3ʳᵈ row) from 2B-CLDCLASS-lidar (left column), from surface (middle column), and difference of 2B-CLDCLASS-lidar and surface at Eureka for 2006-2010.**





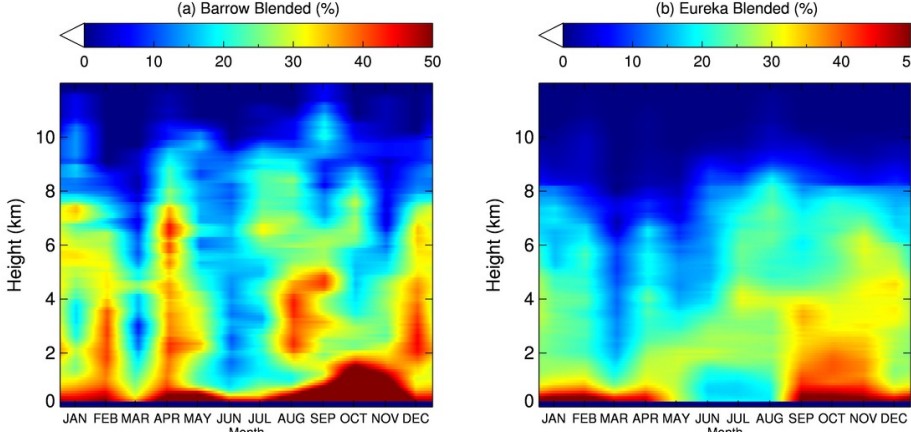

**Figure 10: Blended cloud fraction/frequency vertical distribution at Barrow and Eureka with combined surface and space observations from 2B-GEOPROF-lidar for 2006-2010.**




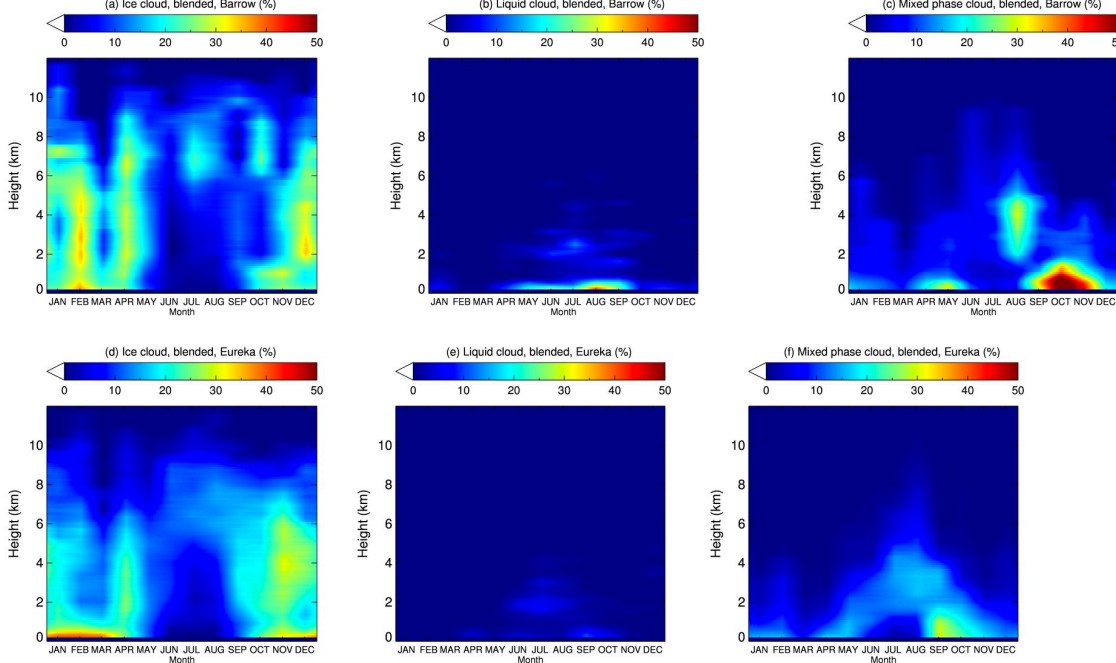

**Figure 11: Blended vertical distributions of (a) ice phase cloud, (b) liquid phase cloud, and (d) mixed-phase cloud at Barrow, and (d) ice phase cloud, (e) liquid phase cloud, and (f) mixed-phase cloud at Eureka from 2B-CLDCLASS-lidar and surface observations for 2006-2010.**

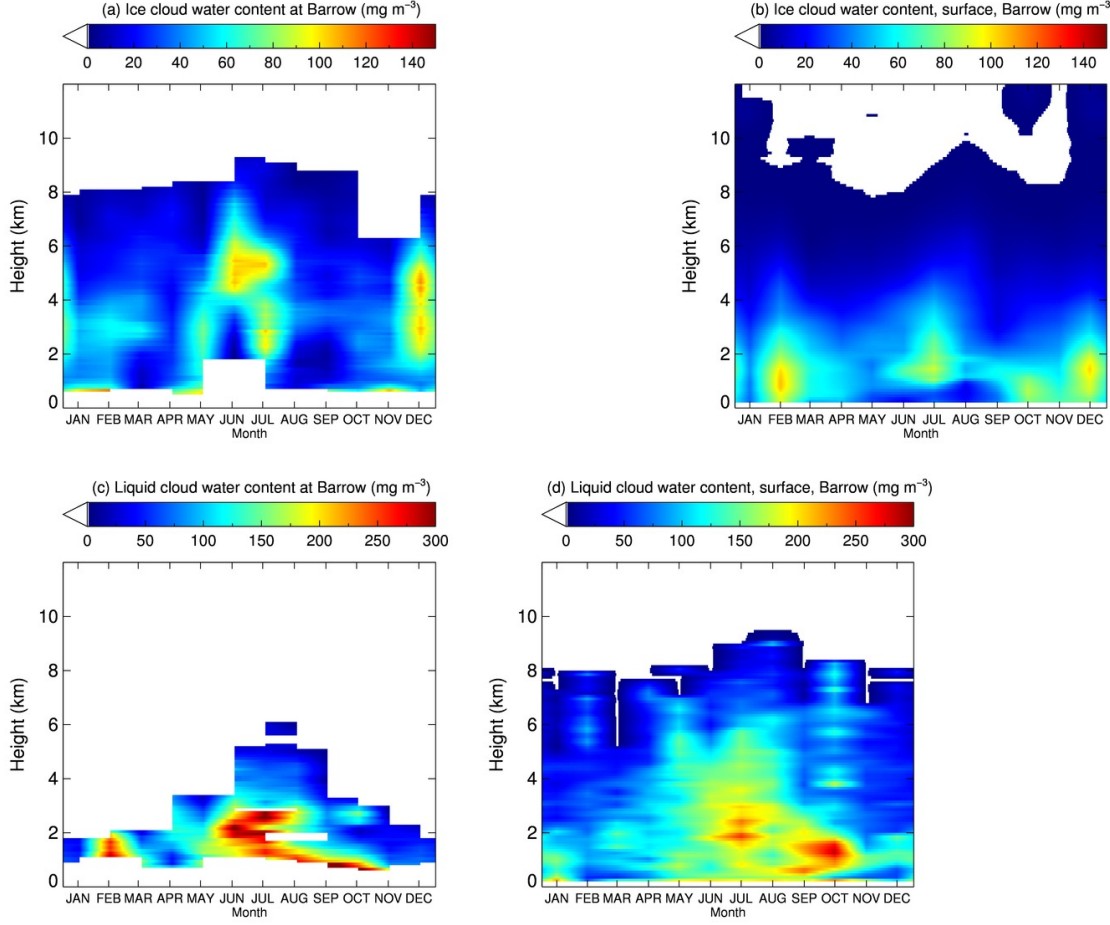

**Figure 12: Vertical Distributions of cloud water content for ice cloud from (a) 2B-CWC-RO, and (b) surface, for liquid cloud from (c) 2B-CWC-RO, and (d) surface at Barrow for 2006-2010.**



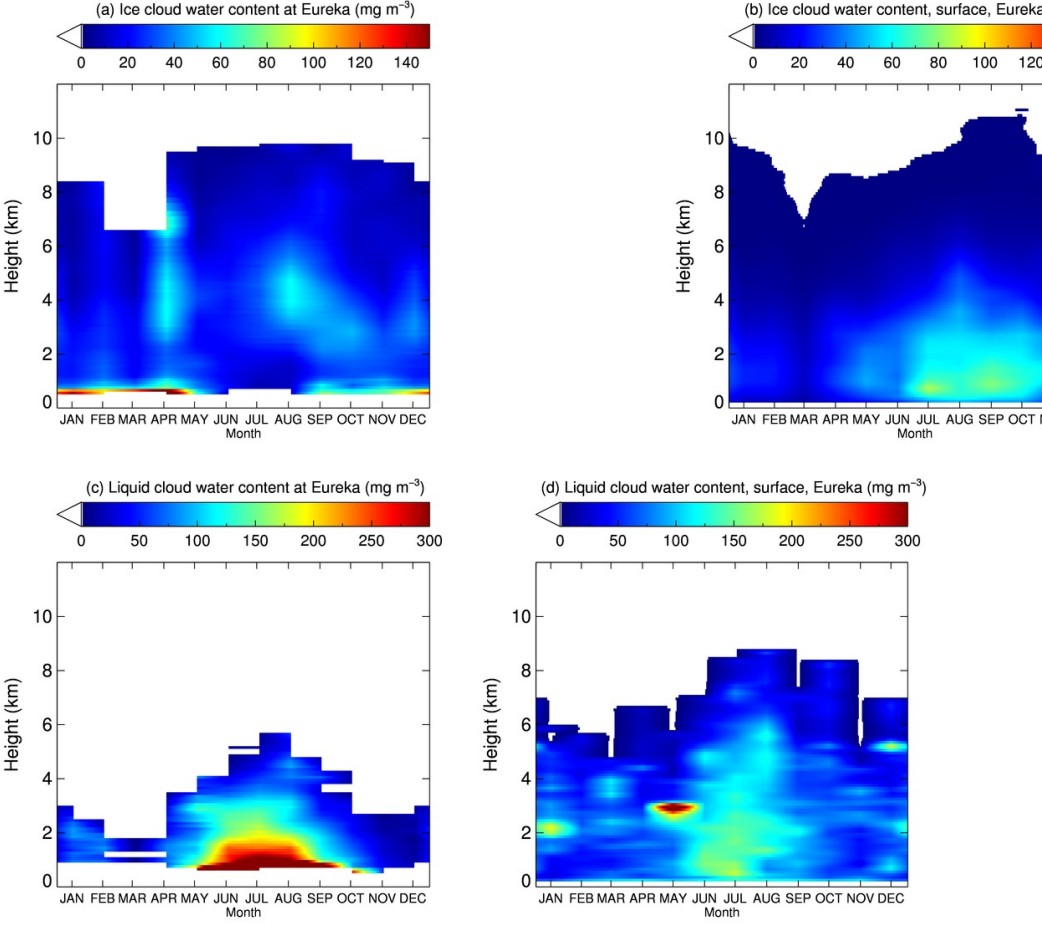

**Figure 13: Same as Figure 12, but for Eureka.**

