# Peer review of "Cloud vertical distribution from combined surface and space radar/lidar observations at two Arctic atmospheric observatories"

_Atmospheric Chemistry and Physics, 2016_

## Referee Comment (RC1) · Anonymous Referee #1 · 31 Jan 2017

The manuscript uses ground- and satellite-based retrievals of cloud fraction, cloud liquid and ice water content and cloud phase profiles from lidar and radar to compare their performance at two Arctic sites: Barrow and Eureka. They propose to merge ground and satellite retrievals of cloud fraction to compensate for their inherent limitations: issues for CloudSat and CALIPSO to detect low-level clouds versus issues for surface based measurements to detect high clouds. I do recommend major revisions as there are some issues with the presentation of the results and the actual content of the conclusions.

1. The method section needs some extensive work, because the explanations are currently confusing and insufficient. I have detailed the problems in the specific comments

below. Are monthly means calculated and used throughout? This is never explicitly said.

2. The detectability issue with CloudSat and CALIPSO for low level clouds is not new, there are already a number of papers that discuss this, e.g. Kay and Gettelman 2009, or Huang et al. (JCLI, 2012, doi: 10.1175/JCLI-D-11-00131.1). The real novelty of this paper is 1) to give an estimate/magnitude to this deficiency and 2) inspect the consequences when looking at the annual cycle of cloud cover in the Arctic. This should be made more prominent.

3. The authors have decided to separate the results from Barrow from Eureka. Why is this? Are the two sites giving different results other than differing climatologies?

4. Although a blended product is a good idea, because of the good performance overall of the surface-based observations (even if less high clouds are detected, the differences with the satellite based observations are small, possibly because of the location and type of clouds). I wonder if such a product is that needed for these two locations. It might be of more use if done for the tropics.

Specific comments:

1. The title is awkward: shouldn't "observations" be "observatories"? or add "sites" at the end.

2. Line 28, page 2: Here, and elsewhere, the authors refer to CloudSat&CALIPSO as "space-based radar-lidar" which makes it quite general when one could imagine that other (future) radars and lidars might have different sensitivities and consequently issues/strengths. If for example the characteristics of the Earthcare mission instruments will be such that they will experience the same problems, then this should be said. Otherwise it would be better in the introduction to say that when referring to "space-based radar-lidar" the authors mean CloudSat and CALIPSO.

Section 2:

Interactive
comment

3. What is the temporal resolution of the profiles, surface and satellite based, when they are compared? Monthly means? Does it mean that the surface profiles are accumulated over a month and then cloud fraction calculated using a cloud mask? Please explain.

4. Throughout the manuscript, please specify whether the lowest levels are identified about the surface or above mean sea level (which presumably is rather close at the two sites? This is not specified).

5. Page 3, line 17: when introducing VFM, please specify which resolution, vertical or horizontal? Depending on which the 1/3, 1 and 5 km refer to, then specify the other resolution. This might help understand the method described on page 4 (see point 10 below)

6. When using GEOPROF, the authors choose the CPR_cloud_mask variable to be above 20 for a range bin to be cloudy. What is the convention in GEOPROF-LIDAR? How does this choice affect the results?

7. Line 20, page 4: here the authors specify that the satellite based profiles are selected if found within 50 km from the sites. Given the narrow swath and polar orbit, how many orbits per month actually fulfill this condition of at least one profile within 50 km? Do "6000 total sample numbers" and "1500 total sample numbers" refer to the total number of profiles?

8. Page 4, lines 21-23: this sentence is confusing, maybe a simple schematic would help visualize what you mean? What is the original vertical resolution of each product?

9. Lines 23-25 page 4: this sentence does make any sense, what is a "cloud case number"? again maybe a schematic would help. Then at the end of the sentence "in a selected time period" refers to a month?

10. Page 4, Lines 25 onward on how the CALIPSO profiles are dealt with: again a schematic might help, as well as a clear explanation of what the horizontal and vertical

resolution of these profiles are, and what it means to combine the 1/3 and 1 or 5 km products. Finally, what is the final vertical resolution of all of the products (CloudSat alone, CALIPSO alone, combined and surface)? Also why use both the 1/3 and 1km combination and the 1/3 and 5 km combination?

11. Page 5, last paragraph of section 2: are the surface products only selected when coincident with an A-train orbit? And, most importantly, are the profiles to be used in section 3 monthly means/accumulations??

12. Figure 1, 2, 7, 9, 10 and 11: the color bar covers 0-50% but from the text cloud fractions exceed this value at low levels it seems. Why not use the full range of available values?

13. How is the "monthly mean total cloud amount" calculated for each instrument? (e.g. line 28, page 6)

14. Figures 4, 5, 6 need to be redone with either thicker lines or (better) in color, to help distinguish between the different lines. It is really hard to read these as they are.

15. Page 7, sentence on lines 3-4: this is awkward, since you've already explained that the surface products were described in Shupe (2007, 2011), why not skip this first sentence and add reference to these two studies in the next sentence.

16. Page 9, line 10: "Major differences" between what? Barrow and Eureka or surface and satellite?

17. Section 3.2: more information is needed: what is the temporal resolution of the combined product? If monthly means, then this is a combination of the monthly means from surface and satellite? Or are these constructed for coincident observations only? Then how are the two products reconciled in term of surface time average vs satellite spatial average? Line 27: "a complete picture of the " monthly "cloud fraction vertical distribution"?

18. Section 3.3: what is the take-home message for this section?

[Figure]

19. Conclusions: the first "primary conclusion" is the direct consequence of the known limitations in the CloudSat (surface clutter/low sensitivity) and the CALIPSO (attenuation) instruments. References to other studies should be given. For the second "primary conclusion", I would be inclined to conclude that surface observations perform well, regardless of cloud altitude. For the third conclusion, I would encourage the authors to discuss a bit more the implications for the annual cycle of the satellite based deficiencies. Finally, although I agree that the blended product is more accurate than surface only observations, I think that the real advantage is if one is to calculate heating rates and/or TOA/surface fluxes, this is where this product might make a difference. This should be discussed.

20. Finally, two papers come to mind to address the very last sentence of the paper, where combined satellite products were used to evaluate cloud impacts in the Arctic in Kay et al (2008) and Kay and Gettelman (2009). The authors might want to mention these results.

Typos

1. Abstract, line 24: remove "annual cycle" after "vertical distribution"

2. Line 24, page 3: "negligible surface above 0.96 km" does not make sense, is "clutter" missing?

3. Line 26, page 6: please add "to" before "penetrate" and "thick" after "optically"

4. Line 34, page 6: replace "the" before "CloudSat" with "that".

5. Line 13, page 7: add "with" before "2B-CLDCLASS-lidar"

6. Page 8, line 21: remove "This" after "Whether"

7. Page 8. Lone 23: "the" instead of "he" before "whole Arctic"

Kay, J. E., and A. Gettelman (2009), Cloud influence on and response to seasonal Arctic sea ice loss, J. Geophys. Res., 114, D18204, doi:10.1029/2009JD011773 Kay,

J. E., T. L'Ecuyer, A. Gettelman, G. Stephens, and C. O'Dell (2008), The contribution of cloud and radiation anomalies to the 2007 Arctic sea ice extent minimum, Geophys. Res. Lett., 35, L08503, doi:10.1029/2008GL033451

---

## Referee Comment (RC2) · A. Devasthale (Referee) · 6 Feb 2017

Abhay Devasthale, SMHI, Norrköping, Sweden                                              2017-02-06

**Review of Liu et al. doi:10.5194/acp-2016-1132, 2017**

While I am a core satellite believer, I do understand and appreciate the importance of in-situ measurements, especially in anchoring space based observations. And there is perhaps no other region in the world where we desperately need more in-situ observations than in the Arctic. Combining these two (space based and in-situ) observing systems is even better. So I really appreciate the work done by the authors in this regard. I have few issues mentioned below that I regard minor in nature, but need to be explained/elaborated. I also had an opportunity to go through the comments posted by the other reviewer and I broadly agree with her/him and I hope the authors will address them as well.

1) The authors discuss a great deal about how they compute vertical cloud fraction, but very little (or almost nothing if I haven't missed anything obvious) about the spatial (and temporal) collocation of space based and in-situ measurements. The impact of uncertainties arising from these issues is not be underestimated, especially when you compare and combined products with different spatial resolutions (even at monthly mean scale). Let's say that you (or CALIPSO team) use 15 CALIOP single shots (1/3 km each, 5x3) to generate 5 km product. What happens when this 5 km product is not centered over Barrow or Eureka and you are inconsistently selecting single shots? Have the authors evaluated few individual cases manually to check what to expect when they merge 1/3, 1 and 5 km data with reference to the station in question?

2) It would be helpful if the authors also provide some physical explanation of the seasonal highs and lows in cloud fractions seen in the results. For example, in the case of Barrow, why is cloud fraction peaking in Feb, Apr and Oct months? Why is there a minimum in Jun and Jul?  This is different from Eureka. Why? Perhaps Shupe et al (2011; 2015) already discuss this, but I think the reader still needs at least a brief description of it to make full sense of the differences you observe from these two observing systems.

3)  In the case of Barrow station, I am bit surprised at the differences in CF between 2B-GEORPFO and 2B-GEORPOF-Lidar in Aug (Fig. 2). When you add CALIOP there seems to be increase in clouds in the free troposphere from 1 to 5 km. Instinctively, I would have thought that, in the free troposphere, CALIOP would add those subvisual or super thin clouds that are missed by CPR, located in the upper troposphere lower stratosphere. Nearly 30-40% more clouds are added by GEOPROF-Lidar compared to GEOPROF in the lower and middle troposphere and it seems that even surface measurements missed these clouds. Even more confusing is the fact that CALIPSO 5 km doesn't show these clouds in Aug. So what is happening here? Part of this discrepancy can be due to the attenuation of CALIOP signal and part of it due to high amount thin clouds in the middle and lower troposphere (Devasthale et al. 2011). But it is difficult to say without further investigations.

4) The authors say that the blended cloud vertical distribution provides a complete picture. But how do we quantitatively know this? After all, we need a third independent reference to make that conclusion.

Reference:

http://onlinelibrary.wiley.com/doi/10.1111/j.1600-0889.2010.00516.x/pdf

---

## Author Comment (AC1) · 8 Apr 2017

The response to the reviewers' comments is in italic.

Anonymous Referee #1

The manuscript uses ground- and satellite-based retrievals of cloud fraction, cloud liquid and ice water content and cloud phase profiles from lidar and radar to compare their performance at two Arctic sites: Barrow and Eureka. They propose to merge ground and satellite retrievals of cloud fraction to compensate for their inherent limitations: issues for CloudSat and CALIPSO to detect low-level clouds versus issues for surface based measurements to detect high clouds. I do recommend major revisions as there are some issues with the presentation of the results and the actual content of the conclusions.

*We appreciate the reviewer's valuable comments. The manuscript becomes better with revisions in response to reviewer's comments and suggestions.*

1. The method section needs some extensive work, because the explanations are currently confusing and insufficient. I have detailed the problems in the specific comments below. Are monthly means calculated and used throughout? This is never explicitly said.

*Changes have been made in the method section in the revised manuscript in response to reviewer's suggestions. Details can be found in the response to reviewer's specific comments below. Monthly means are calculated and used throughout, and this is specified in the revised manuscript.*

2. The detectability issue with CloudSat and CALIPSO for low level clouds is not new, there are already a number of papers that discuss this, e.g. Kay and Gettelman 2009, or Huang et al. (JCLI, 2012, doi: 10.1175/JCLI-D-11-00131.1). The real novelty of this paper is 1) to give an estimate/magnitude to this deficiency and 2) inspect the consequences when looking at the annual cycle of cloud cover in the Arctic. This should be made more prominent.

*The references the reviewers suggested have been added in the revised manuscript with correspondent discussion. The reviewer also summarized the novelty of our work well, and we highly appreciated that and have included that in the revised manuscript.*

3. The authors have decided to separate the results from Barrow from Eureka. Why is this? Are the two sites giving different results other than differing climatologies?

*We actually spent quite some time figuring out the best way to present the results, either separating by different physical parameters, e.g. cloud amount, cloud phase, and cloud water content, or by different locations, e.g. Barrow and Eureka. We then decided to go with the latter for clearer presentation. The climatologies at these two sites are not the same, so we do not think the content are redundant.*

4. Although a blended product is a good idea, because of the good performance overall of the surface-based observations (even if less high clouds are detected, the differences with the satellite based observations are small, possibly because of the location and type of clouds). I wonder if such a product is that needed for these two locations. It might be of more use if done for the tropics.

*We totally agree with the reviewer that such a blended product might be more useful in the tropics. We would like to argue that such products may be as valuable in the polar regions as*

*they are in the tropics because of the ubiquitous low-level clouds in the polar regions, and lack of detection capability from CloudSat and CALIPSO. Such discussions have been added in the revised manuscript.*

Specific comments:

1. The title is awkward: shouldn't "observations" be "observatories"? or add "sites" at the end.

*In the title, "observations" was changed to "observatories".*

2. Line 28, page 2: Here, and elsewhere, the authors refer to CloudSat&CALIPSO as "space-based radar-lidar" which makes it quite general when one could imagine that other (future) radars and lidars might have different sensitivities and consequently issues/ strengths. If for example the characteristics of the Earthcare mission instruments will be such that they will experience the same problems, then this should be said. Otherwise it would be better in the introduction to say that when referring to "space-based radar-lidar" the authors mean CloudSat and CALIPSO.

*Responding to reviewer's comment, the following text has been added in the revised manuscript. "Space-based radar and lidar in this paper refer to existing instruments, i.e. Cloud Profiling Radar (CPR) onboard the CloudSat and the Cloud-Aerosol LIdar with Orthogonal Polarization (CALIOP) onboard the Cloud–Aerosol lidar and Infrared Pathfinder Satellite Observation (CALIPSO). However, the conclusions will likely be valid for the space-based radar and lidar instruments in the foreseeable future, i.e. the ATmospheric backscatter LIDar (ATLID), and the CPR onboard the EarthCARE mission (Heliere et al. 2007)."*

Section 2:
3. What is the temporal resolution of the profiles, surface and satellite based, when they are compared? Monthly means? Does it mean that the surface profiles are accumulated over a month and then cloud fraction calculated using a cloud mask? Please explain.

*The temporal resolution in the comparison is monthly. All surface profiles in a month are accumulated for calculation of monthly means. This has been added in the revised manuscript.*

4. Throughout the manuscript, please specify whether the lowest levels are identified about the surface or above mean sea level (which presumably is rather close at the two sites? This is not specified).

*The lowest levels are identified above the mean sea level. This has been added in the revised manuscript, "Monthly means are calculated for both surface observations and for the space-based sensors. All heights are above the mean sea level. All surface profiles in a month are accumulated for calculation of monthly means.", and "The vertical resolution of the calculated monthly means is interpolated to 100 m to be consistent with those from surface observations." In the last paragraph of section 2.*

5. Page 3, line 17: when introducing VFM, please specify which resolution, vertical or horizontal? Depending on which the 1/3, 1 and 5 km refer to, then specify the other resolution. This might help understand the method described on page 4 (see point 10 below)

*The following text has been added in the revised manuscript, "The Vertical Feature Mask (VFM) from CALIPSO's CALIOP provides cloud vertical distribution in up to 10 vertical layers at 5 km*

*and 1 km horizontal resolutions, and up to 5 vertical layers at 1/3 km horizontal resolution (Vaughan et al. 2009). The vertical resolution is 30 m below 8.2 km, and 60 m between 8.2 and 20.2 km. A Selective Iterated BoundarY Location (SIBYL) scheme is applied to detect all features within a given scene. Strongly scatter features, e.g. stratus clouds, can be identified in a single laser pulse, with the 1/3 km horizontal resolution, and these features are then removed in order to detect any surrounding aerosol layers. Weakly scattering features, e.g. thin cirrus clouds, are detected with the average of several laser pulses, e.g. 5 km horizontal resolution, for higher signal-to-noise ratio (Vaughan et al. 2005). Compared to the 1 km resolution data, the 5 km resolution product can identify weaker cloud features using an iterative multi-resolution averaging scheme (Vaughan et al. 2009). Combination of the cloud layer products at 5 km and 1/3 km provides a complete vertical distribution of clouds from CALIPSO (Vaughan et al. 2009, Vaughan et al. 2005).".*

6. When using GEOPROF, the authors choose the CPR_cloud_mask variable to be above 20 for a range bin to be cloudy. What is the convention in GEOPROF-LIDAR? How does this choice affect the results?

*The threshold in the GEOPROF-LIDAR is also 20. In the revised manuscript, we added, "This threshold is the same as that used in the 2B-GEOPROF-LIDAR (Mace et al. 2009, Mace et al. 2009). A false positive detection of 5% is estimated with this threshold in the 2B-GEOPROF-LIDAR (Mace et al. 2009)" The impact of the choice on the results is beyond the scope of this study.*

7. Line 20, page 4: here the authors specify that the satellite based profiles are selected if found within 50 km from the sites. Given the narrow swath and polar orbit, how many orbits per month actually fulfill this condition of at least one profile within 50 km? Do "6000 total sample numbers" and "1500 total sample numbers" refer to the total number of profiles?

*The text in the revised manuscript has been changed as "The monthly mean sample number of the satellite sensors is a function of latitude in the Arctic, with the fewest at 60° N, gradually increasing to a maximum around 80° N (Liu 2015). Both factors are reflected in the large number of samples at Eureka, with over 6000 total samples per month from June 2006 to December 2010 at Eureka, and around 1500 total samples at Barrow per month from middle February 2008 to December 2010.".*

8. Page 4, lines 21-23: this sentence is confusing, maybe a simple schematic would help visualize what you mean? What is the original vertical resolution of each product?

*The vertical resolution is 30 m below 8.2 km, and 60 m between 8.2 and 20.2 km. The vertical resolution of 2B-GEOPROF and 2B-GEOPROF-lidar are at 240 m. We added this information in the revised manuscript.*

*A schematic would be great. But we did not figure out a way to make a simple schematic. So, we re-wrote the description to calculate the mean cloud vertical distribution.*

9. Lines 23-25 page 4: this sentence does make any sense, what is a "cloud case number"? again maybe a schematic would help. Then at the end of the sentence "in a selected time period" refers to a month?

*A schematic would be great. But we did not figure out a way to make a simple schematic. So, we re-wrote the description to calculate the mean cloud vertical distribution as the following in the revised manuscript "Vertical profiles of all these products within 50 km of the two Arctic atmospheric observation sites, Barrow and Eureka, are extracted and archived. The cloud fraction vertical distribution at a resolution of 30 m is calculated as follows. The mean cloud fraction at each vertical level is calculated as the ratio of number of profiles with cloud detected at this vertical level to the total profile numbers. The cloud vertical distribution from CALIPSO at 1/3 km and 5 km are calculated first, then combined as the mean of the cloud fractions from CALIPSO 1/3 km and 5 km at each vertical level. This combined product is referred as CALIPSO 5 km, provides a complete vertical distribution of clouds from CALIPSO, and is shown in section 3. To compare, the vertical profiles of cloud fraction from CALIPSO at 1/3 km and 1 km are also combined, and shown in section 3. The combined product is referred as CALIPSO 1 km. For cloud microphysical property vertical distribution, the mean cloud phase frequency at each vertical level is calculated as the ratio of numbers of profiles with each phase to the total profile numbers. Mean cloud water content for ice (liquid) phase at each vertical level is calculated as the mean values of water content from all available ice (liquid) cloud retrievals at that level. For deriving these statistics, ice in any type of cloud (ice and mixed phase) is included, while liquid in any type of cloud (liquid and mixed phase) is included. After this step, the vertical resolution of all products is 30 m. Total cloud (ice cloud, liquid cloud, mixed phase cloud) amounts are also calculated, as the ratio of number of profiles with cloud (ice cloud, liquid cloud, mixed phase cloud) detected in any layer to the total number of profiles".*

10. Page 4, Lines 25 onward on how the CALIPSO profiles are dealt with: again a schematic might help, as well as a clear explanation of what the horizontal and vertical resolution of these profiles are, and what it means to combine the 1/3 and 1 or 5 km products. Finally, what is the final vertical resolution of all of the products (CloudSat alone, CALIPSO alone, combined and surface)? Also why use both the 1/3 and 1km combination and the 1/3 and 5 km combination?

*Please see response to comment #4 and #9. As stated in the manuscript, it would be meaningful to see how combined 1/3 km and 1 km compares to combined 1/3 km and 5 km. The comparison of combined 1/3 km and 5 km shows more complete description, as we expected.*

*As in the response to comment #9, the vertical resolution is 30 m. We then interpolated to 100 m to be consistent with and compared to those from surface observations. These have been added in the revised manuscript.*

11. Page 5, last paragraph of section 2: are the surface products only selected when coincident with an A-train orbit? And, most importantly, are the profiles to be used in section 3 monthly means/accumulations??

*All surface profiles in a month are included in the monthly mean calculation. This is specified in the revised manuscript. "Monthly means are calculated for both surface observations and for the space-based sensors. All heights are above the mean sea level. All surface profiles in a month are accumulated for calculation of monthly means."*

12. Figure 1, 2, 7, 9, 10 and 11: the color bar covers 0-50% but from the text cloud fractions exceed this value at low levels it seems. Why not use the full range of available values?

*Figures 1,2,7,9,10 have been updated in the revised manuscript. The color range extends to 0-80% for Barrow, and 0-60% for Eureka. We also tried extending to 0-100% for both stations, and the details in the figures were not shown as well.*

13. How is the "monthly mean total cloud amount" calculated for each instrument? (e.g. line 28, page 6)

*The following text has been added in the revised manuscript "Total cloud (ice cloud, liquid cloud, mixed phase cloud) amounts are also calculated, as the ratio of number of profiles with cloud (ice cloud, liquid cloud, mixed phase cloud) detected in any layer to the total number of profiles".*

14. Figures 4, 5, 6 need to be redone with either thicker lines or (better) in color, to help distinguish between the different lines. It is really hard to read these as they are.

*Figure 4 and 5 have been updated with lines in color in the revised manuscript. We think the lines in Figure 6 are clear, so we did not update Figure 6.*

15. Page 7, sentence on lines 3-4: this is awkward, since you've already explained that the surface products were described in Shupe (2007, 2011), why not skip this first sentence and add reference to these two studies in the next sentence.

*Revised as the reviewer suggested.*

16. Page 9, line 10: "Major differences" between what? Barrow and Eureka or surface and satellite?

*This paragraph has been revised as the following, "Vertical distributions of ice cloud, liquid cloud, and mixed phase cloud at Eureka from space-based observations show similar patterns above 1 km as those from surface observations (Figure 9). The major differences between surface and space-based observations in the cloud vertical distributions at Eureka (Figure 8d, 8e, 8f, and Figure 9) are similar to those at Barrow (Figure 7, Figure 8a, 8b, and 8c). Major differences between surface and space-based observations include: much less ice and mixed phase cloud in the lowest 1 km from space-based observations; greater liquid cloud, and mixed phase cloud above 2 km in the vertical distributions and annual mean of vertical distributions from space-based observations (Figure not shown); comparable monthly mean total cloud amount, higher ice cloud monthly means, lower liquid cloud monthly means, and higher mixed phase cloud monthly means from surface observations relative to space-based observations. In additions, both satellite and surface observations reveal a key difference to the annual cycles of clouds at Eureka versus Barrow.  While both sites have a similar annual cycle of ice cloud occurrence with a relative decrease in summer (Figure 8a, and 8d), there are less frequent liquid-containing clouds at Eureka with the annual maximum of these generally shifted to the autumn. These relative annual cycles explain the key differences in total cloud occurrence fraction over the annual cycle and are explained by generally colder and drier conditions in Eureka relative to Barrow (e.g., Shupe 2011).".*

17. Section 3.2: more information is needed: what is the temporal resolution of the combined product? If monthly means, then this is a combination of the monthly means from surface and satellite? Or are these constructed for coincident observations only? Then how are the two products reconciled in term of surface time average vs satellite spatial average? Line 27: "a complete picture of the " monthly "cloud fraction vertical distribution"?

*The blended product is in monthly means. Line 27 has been revised as the reviewer suggested.*

18. Section 3.3: what is the take-home message for this section?

*The following text has been added in the revised manuscript, "These comparisons indicate that liquid water content monthly means from space-based and surface observations show similar annual evolution with noticeable magnitude differences. The ice water content monthly means from space and surface observations share little similarities in annual evolution or magnitude. Further investigation of these differences is warranted in order to combine these products for a complete vertical distribution of cloud water content".*

19. Conclusions: the first "primary conclusion" is the direct consequence of the known limitations in the CloudSat (surface clutter/low sensitivity) and the CALIPSO (attenuation) instruments. References to other studies should be given. For the second "primary conclusion", I would be inclined to conclude that surface observations perform well, regardless of cloud altitude. For the third conclusion, I would encourage the authors to discuss a bit more the implications for the annual cycle of the satellite based deficiencies. Finally, although I agree that the blended product is more accurate than surface only observations, I think that the real advantage is if one is to calculate heating rates and/or TOA/surface fluxes, this is where this product might make a difference. This should be discussed.

*All the suggestions are well received, and correspondent discussions have been added in the revised manuscript as suggested by the reviewer.  In each of the primary conclusions an additional sentence or more has been added to better capture implications and context.*

20. Finally, two papers come to mind to address the very last sentence of the paper, where combined satellite products were used to evaluate cloud impacts in the Arctic in Kay et al (2008) and Kay and Gettelman (2009). The authors might want to mention these results.

*We agree. Kay et al. (2008) and Kay and Gettelman (2009) used combined satellite products. The last sentence of the manuscript suggests that we need combine surface-based and satellite products, in addition to combined satellite products. However, we appreciate the suggestions, and both references have been included in the revised manuscript.*

Typos

1. Abstract, line 24: remove "annual cycle" after "vertical distribution"

2. Line 24, page 3: "negligible surface above 0.96 km" does not make sense, is "clutter" missing?

3. Line 26, page 6: please add "to" before "penetrate" and "thick" after "optically"

4. Line 34, page 6: replace "the" before "CloudSat" with "that".

5. Line 13, page 7: add "with" before "2B-CLDCLASS-lidar"

6. Page 8, line 21: remove "This" after "Whether"

7. Page 8. Lone 23: "the" instead of "he" before "whole Arctic"

Kay, J. E., and A. Gettelman (2009), Cloud influence on and response to seasonal Arctic sea ice loss, J. Geophys. Res., 114, D18204, doi:10.1029/2009JD011773

Kay, J. E., T. L'Ecuyer, A. Gettelman, G. Stephens, and C. O'Dell (2008), The contribution of cloud and radiation anomalies to the 2007 Arctic sea ice extent minimum, Geophys. Res. Lett., 35, L08503, doi:10.1029/2008GL033451

*All the typos have been corrected. Both references have been added in the revised manuscript.*

---

## Author Comment (AC2) · 8 Apr 2017

The response to the reviewers' comments is in italic.

Abhay Devasthale, SMHI, Norrköping, Sweden 2017-02-06

**Review of Liu et al. doi:10.5194/acp-2016-1132, 2017**

While I am a core satellite believer, I do understand and appreciate the importance of in-situ measurements, especially in anchoring space based observations. And there is perhaps no other region in the world where we desperately need more in-situ observations than in the Arctic. Combining these two (space based and in-situ) observing systems is even better. So I really appreciate the work done by the authors in this regard. I have few issues mentioned below that I regard minor in nature, but need to be explained/elaborated. I also had an opportunity to go through the comments posted by the other reviewer and I broadly agree with her/him and I hope the authors will address them as well.

*We appreciate Dr. Devasthale's valuable comments. The manuscript becomes better with revisions in response to reviewer's comments and suggestions. We have responded to other reviewer's comments point by point, and made correspondent revisions in the revised manuscript.*

1) The authors discuss a great deal about how they compute vertical cloud fraction, but very little (or almost nothing if I haven't missed anything obvious) about the spatial (and temporal) collocation of space based and in-situ measurements. The impact of uncertainties arising from these issues is not be underestimated, especially when you compare and combined products with different spatial resolutions (even at monthly mean scale). Let's say that you (or CALIPSO team) use 15 CALIOP single shots (1/3 km each, 5x3) to generate 5 km product. What happens when this 5 km product is not centered over Barrow or Eureka and you are inconsistently selecting single shots? Have the authors evaluated few individual cases manually to check what to expect when they merge 1/3, 1 and 5 km data with reference to the station in question?

*We totally agree with the reviewer's comments, and thank for his insight. These issues, e.g. cloud frequency from surface observations v.s. spatial coverage from space-based observations, different spatial resolutions, viewing angles, vertical resolution among satellite products, all contribute to the shown differences in this manuscript. By using long-term observations, e.g. over 4 years at Eureka and over 2 years at Barrow (all data we have right now), we believe the temporal and spatial average would mitigate these issues. When longer term data from both surface-based and space-based are available, it is worth to revisit this, and see how the differences would change.*

*Inspired by the reviewer's comments, we add a paragraph in the "Conclusion" as the following, "Cloud frequency from surface is calculated in the temporal domain, while the cloud fraction from space-based observations is calculated in the spatial domain although near the surface sites. Differences in spatial resolution, viewing angles, vertical resolution, instrument sensitivity to clouds and retrieval algorithms may all contribute to the differences in the cloud vertical distributions from different instruments. Long-term averages of products may mitigate the impacts of some of these factors. Causes of the remaining differences are worth further investigation.*
*"*

2) It would be helpful if the authors also provide some physical explanation of the seasonal highs and lows in cloud fractions seen in the results. For example, in the case of Barrow, why is cloud

fraction peaking in Feb, Apr and Oct months? Why is there a minimum in Jun and Jul? This is different from Eureka. Why? Perhaps Shupe et al (2011; 2015) already discuss this, but I think the reader still needs at least a brief description of it to make full sense of the differences you observe from these two observing systems.

*A short description of the difference between Barrow and Eureka has been added to the end of Section 3.1.2. This explanation also links to a more detailed discussion of the matter in Shupe (2011). The discussion is "In additions, both satellite and surface observations reveal a key difference to the annual cycles of clouds at Eureka versus Barrow. While both sites have a similar annual cycle of ice cloud occurrence with a relative decrease in summer (Figure 8a, and 8d), there are less frequent liquid-containing clouds at Eureka with the annual maximum of these generally shifted to the autumn. These relative annual cycles explain the key differences in total cloud occurrence fraction over the annual cycle and are explained by generally colder and drier conditions in Eureka relative to Barrow (e.g., Shupe 2011)".*

3) In the case of Barrow station, I am bit surprised at the differences in CF between 2B-GEORPFO and 2B-GEORPOF-Lidar in Aug (Fig. 2). When you add CALIOP there seems to be increase in clouds in the free troposphere from 1 to 5 km. Instinctively, I would have thought that, in the free troposphere, CALIOP would add those subvisual or super thin clouds that are missed by CPR, located in the upper troposphere lower stratosphere. Nearly 30-40% more clouds are added by GEOPROF-Lidar compared to GEOPROF in the lower and middle troposphere and it seems that even surface measurements missed these clouds. Even more confusing is the fact that CALIPSO 5 km doesn't show these clouds in Aug. So what is happening here? Part of this discrepancy can be due to the attenuation of CALIOP signal and part of it due to high amount thin clouds in the middle and lower troposphere (Devasthale et al. 2011). But it is difficult to say without further investigations.

*I agree with the reviewer's comment. The GEOPROF-Lidar has higher values than the sum of those from 2B-GEOPROF and CALIPSO 5 km in August at Barrow. The reviewer gave some possible causes, and we appreciated that and have included such discussion in the revised manuscript. However, it is still unclear why the 2B-GEOPROF-lidar has higher values than the sum of those from 2B-GEOPROF and CALIPSO 5 km. Though finding the causes is beyond the scope of this study, it is worth further investigation in future work. The following discussion has been added in the revised manuscript.*
*"It is worth pointing out that the 2B-GEOPROF-LIDAR shows higher cloud amount values from 1 km to 5 km in the troposphere than the sum of cloud amounts from 2B-GEOPROF and CALIPSO 5 km. The differences can be partially attributed to the attenuation of CALIOP signal and high amount thin clouds in the middle and lower troposphere (Devasthale et al. 2011). Though attribution investigation is beyond the scope of this study, it is worth further investigation in future studies".*

4) The authors say that the blended cloud vertical distribution provides a complete picture. But how do we quantitatively know this? After all, we need a third independent reference to make that conclusion.

*We totally agree. A 3-D cloud distribution product would be ideal with known uncertainties. However, such a product does not exist, and probably will not be available in the near future. So, in my humble opinion, we need to work hard on getting the uncertainties of the existing products, and hopefully merging them for better quality. That is the motivation of this study.*

---

## Author Response (AR2)

Dear Dr. Garrett,

Thanks very much for your evaluation of our revised manuscript.

We have made correspondent revisions according to your comments.

In reply to your comments that "There are two points I would like to see addressed. First is that the English is at times idiosyncratic and could benefit from some rewriting from a native English speaker." , I asked an editor here in Cooperative Institute for Meteorological Satellite Studies, Ms. Leanne Avila, to read , correct grammatical errors, and edit the revised manuscript.

In regarding to your second point, i.e. "ACP is not primarily a technical journal, so to be publishable the work needs to be placed in a broader scientific context. There is some very interesting information on the seasonal cycles of clouds at Barrow and Eureka that is described in Section 3 but not mentioned in either the abstract or conclusions. Nor are the results contrasted with prior ground and space based studies into the seasonal cycles of clouds in the Arctic and at Barrow and Eureka. As reviewer 2 pointed out, there is a broad interest in arctic clouds, so greater emphasis could be placed on these points.", we have added text in the sections of abstract and conclusion. In the abstract, we added, "Cloud annual cycles show similar evolutions in total cloud fraction and ice cloud fraction, and lower liquid-containing cloud fraction at Eureka than at Barrow; the differences can be attributed to the generally colder and drier conditions in Eureka relative to Barrow". In the conclusion, we added "Annual cycles of the total cloud fraction at Barrow and Eureka show a similar evolution, with highest values in autumn, e.g. September and October, and local minimum values in summer, e.g. June and July, and with generally higher monthly cloud fractions at Barrow except in January and February. Annual cycles of ice clouds at both sites also have a similar evolution with a relative decrease in summer, and show similar magnitude; liquid-containing clouds at Eureka show lower values than those at Barrow, and its maximum generally shifts to the autumn relative to that at Barrow. These similarities and differences in annual cycles explain the key differences in the total cloud fractions, and can be attributed to the generally colder and drier conditions in Eureka relative to Barrow (e.g., Shupe 2011)".

Thanks very much for your evaluation of our manuscript.

Sincerely,

Yinghui Liu

The response to the reviewers' comments is in italic.

Anonymous Referee #1

The manuscript uses ground- and satellite-based retrievals of cloud fraction, cloud liquid and ice water content and cloud phase profiles from lidar and radar to compare their performance at two Arctic sites: Barrow and Eureka. They propose to merge ground and satellite retrievals of cloud fraction to compensate for their inherent limitations: issues for CloudSat and CALIPSO to detect low-level clouds versus issues for surface based measurements to detect high clouds. I do recommend major revisions as there are some issues with the presentation of the results and the actual content of the conclusions.

*We appreciate the reviewer's valuable comments. The manuscript becomes better with revisions in response to reviewer's comments and suggestions.*

1. The method section needs some extensive work, because the explanations are currently confusing and insufficient. I have detailed the problems in the specific comments below. Are monthly means calculated and used throughout? This is never explicitly said.

*Changes have been made in the method section in the revised manuscript in response to reviewer's suggestions. Details can be found in the response to reviewer's specific comments below. Monthly means are calculated and used throughout, and this is specified in the revised manuscript.*

2. The detectability issue with CloudSat and CALIPSO for low level clouds is not new, there are already a number of papers that discuss this, e.g. Kay and Gettelman 2009, or Huang et al. (JCLI, 2012, doi: 10.1175/JCLI-D-11-00131.1). The real novelty of this paper is 1) to give an estimate/magnitude to this deficiency and 2) inspect the consequences when looking at the annual cycle of cloud cover in the Arctic. This should be made more prominent.

*The references the reviewers suggested have been added in the revised manuscript with correspondent discussion. The reviewer also summarized the novelty of our work well, and we highly appreciated that and have included that in the revised manuscript.*

3. The authors have decided to separate the results from Barrow from Eureka. Why is this? Are the two sites giving different results other than differing climatologies?

*We actually spent quite some time figuring out the best way to present the results, either separating by different physical parameters, e.g. cloud amount, cloud phase, and cloud water content, or by different locations, e.g. Barrow and Eureka. We then decided to go with the latter for clearer presentation. The climatologies at these two sites are not the same, so we do not think the content are redundant.*

4. Although a blended product is a good idea, because of the good performance overall of the surface-based observations (even if less high clouds are detected, the differences with the satellite based observations are small, possibly because of the location and type of clouds). I wonder if such a product is that needed for these two locations. It might be of more use if done for the tropics.

*We totally agree with the reviewer that such a blended product might be more useful in the tropics. We would like to argue that such products may be as valuable in the polar regions as*

*they are in the tropics because of the ubiquitous low-level clouds in the polar regions, and lack of detection capability from CloudSat and CALIPSO. Such discussions have been added in the revised manuscript.*

Specific comments:

1. The title is awkward: shouldn't "observations" be "observatories"? or add "sites" at the end.

*In the title, "observations" was changed to "observatories".*

2. Line 28, page 2: Here, and elsewhere, the authors refer to CloudSat&CALIPSO as "space-based radar-lidar" which makes it quite general when one could imagine that other (future) radars and lidars might have different sensitivities and consequently issues/ strengths. If for example the characteristics of the Earthcare mission instruments will be such that they will experience the same problems, then this should be said. Otherwise it would be better in the introduction to say that when referring to "space-based radar-lidar" the authors mean CloudSat and CALIPSO.

*Responding to reviewer's comment, the following text has been added in the revised manuscript. "Space-based radar and lidar in this paper refer to existing instruments, i.e. Cloud Profiling Radar (CPR) onboard the CloudSat and the Cloud-Aerosol LIdar with Orthogonal Polarization (CALIOP) onboard the Cloud–Aerosol lidar and Infrared Pathfinder Satellite Observation (CALIPSO). However, the conclusions will likely be valid for the space-based radar and lidar instruments in the foreseeable future, i.e. the ATmospheric backscatter LIDar (ATLID), and the CPR onboard the EarthCARE mission (Heliere et al. 2007)."*

Section 2:
3. What is the temporal resolution of the profiles, surface and satellite based, when they are compared? Monthly means? Does it mean that the surface profiles are accumulated over a month and then cloud fraction calculated using a cloud mask? Please explain.

*The temporal resolution in the comparison is monthly. All surface profiles in a month are accumulated for calculation of monthly means. This has been added in the revised manuscript.*

4. Throughout the manuscript, please specify whether the lowest levels are identified about the surface or above mean sea level (which presumably is rather close at the two sites? This is not specified).

*The lowest levels are identified above the mean sea level. This has been added in the revised manuscript, "Monthly means are calculated for both surface observations and for the space-based sensors. All heights are above the mean sea level. All surface profiles in a month are accumulated for calculation of monthly means.", and "The vertical resolution of the calculated monthly means is interpolated to 100 m to be consistent with those from surface observations." In the last paragraph of section 2.*

5. Page 3, line 17: when introducing VFM, please specify which resolution, vertical or horizontal? Depending on which the 1/3, 1 and 5 km refer to, then specify the other resolution. This might help understand the method described on page 4 (see point 10 below)

*The following text has been added in the revised manuscript, "The Vertical Feature Mask (VFM) from CALIPSO's CALIOP provides cloud vertical distribution in up to 10 vertical layers at 5 km*

*and 1 km horizontal resolutions, and up to 5 vertical layers at 1/3 km horizontal resolution (Vaughan et al. 2009). The vertical resolution is 30 m below 8.2 km, and 60 m between 8.2 and 20.2 km. A Selective Iterated BoundarY Location (SIBYL) scheme is applied to detect all features within a given scene. Strongly scatter features, e.g. stratus clouds, can be identified in a single laser pulse, with the 1/3 km horizontal resolution, and these features are then removed in order to detect any surrounding aerosol layers. Weakly scattering features, e.g. thin cirrus clouds, are detected with the average of several laser pulses, e.g. 5 km horizontal resolution, for higher signal-to-noise ratio (Vaughan et al. 2005). Compared to the 1 km resolution data, the 5 km resolution product can identify weaker cloud features using an iterative multi-resolution averaging scheme (Vaughan et al. 2009). Combination of the cloud layer products at 5 km and 1/3 km provides a complete vertical distribution of clouds from CALIPSO (Vaughan et al. 2009, Vaughan et al. 2005).".*

6. When using GEOPROF, the authors choose the CPR_cloud_mask variable to be above 20 for a range bin to be cloudy. What is the convention in GEOPROF-LIDAR? How does this choice affect the results?

*The threshold in the GEOPROF-LIDAR is also 20. In the revised manuscript, we added, "This threshold is the same as that used in the 2B-GEOPROF-LIDAR (Mace et al. 2009, Mace et al. 2009). A false positive detection of 5% is estimated with this threshold in the 2B-GEOPROF-LIDAR (Mace et al. 2009)" The impact of the choice on the results is beyond the scope of this study.*

7. Line 20, page 4: here the authors specify that the satellite based profiles are selected if found within 50 km from the sites. Given the narrow swath and polar orbit, how many orbits per month actually fulfill this condition of at least one profile within 50 km? Do "6000 total sample numbers" and "1500 total sample numbers" refer to the total number of profiles?

*The text in the revised manuscript has been changed as "The monthly mean sample number of the satellite sensors is a function of latitude in the Arctic, with the fewest at 60° N, gradually increasing to a maximum around 80° N (Liu 2015). Both factors are reflected in the large number of samples at Eureka, with over 6000 total samples per month from June 2006 to December 2010 at Eureka, and around 1500 total samples at Barrow per month from middle February 2008 to December 2010.".*

8. Page 4, lines 21-23: this sentence is confusing, maybe a simple schematic would help visualize what you mean? What is the original vertical resolution of each product?

*The vertical resolution is 30 m below 8.2 km, and 60 m between 8.2 and 20.2 km. The vertical resolution of 2B-GEOPROF and 2B-GEOPROF-lidar are at 240 m. We added this information in the revised manuscript.*

*A schematic would be great. But we did not figure out a way to make a simple schematic. So, we re-wrote the description to calculate the mean cloud vertical distribution.*

9. Lines 23-25 page 4: this sentence does make any sense, what is a "cloud case number"? again maybe a schematic would help. Then at the end of the sentence "in a selected time period" refers to a month?

*A schematic would be great. But we did not figure out a way to make a simple schematic. So, we re-wrote the description to calculate the mean cloud vertical distribution as the following in the revised manuscript "Vertical profiles of all these products within 50 km of the two Arctic atmospheric observation sites, Barrow and Eureka, are extracted and archived. The cloud fraction vertical distribution at a resolution of 30 m is calculated as follows. The mean cloud fraction at each vertical level is calculated as the ratio of number of profiles with cloud detected at this vertical level to the total profile numbers. The cloud vertical distribution from CALIPSO at 1/3 km and 5 km are calculated first, then combined as the mean of the cloud fractions from CALIPSO 1/3 km and 5 km at each vertical level. This combined product is referred as CALIPSO 5 km, provides a complete vertical distribution of clouds from CALIPSO, and is shown in section 3. To compare, the vertical profiles of cloud fraction from CALIPSO at 1/3 km and 1 km are also combined, and shown in section 3. The combined product is referred as CALIPSO 1 km. For cloud microphysical property vertical distribution, the mean cloud phase frequency at each vertical level is calculated as the ratio of numbers of profiles with each phase to the total profile numbers. Mean cloud water content for ice (liquid) phase at each vertical level is calculated as the mean values of water content from all available ice (liquid) cloud retrievals at that level. For deriving these statistics, ice in any type of cloud (ice and mixed phase) is included, while liquid in any type of cloud (liquid and mixed phase) is included. After this step, the vertical resolution of all products is 30 m. Total cloud (ice cloud, liquid cloud, mixed phase cloud) amounts are also calculated, as the ratio of number of profiles with cloud (ice cloud, liquid cloud, mixed phase cloud) detected in any layer to the total number of profiles".*

10. Page 4, Lines 25 onward on how the CALIPSO profiles are dealt with: again a schematic might help, as well as a clear explanation of what the horizontal and vertical resolution of these profiles are, and what it means to combine the 1/3 and 1 or 5 km products. Finally, what is the final vertical resolution of all of the products (CloudSat alone, CALIPSO alone, combined and surface)? Also why use both the 1/3 and 1km combination and the 1/3 and 5 km combination?

*Please see response to comment #4 and #9. As stated in the manuscript, it would be meaningful to see how combined 1/3 km and 1 km compares to combined 1/3 km and 5 km. The comparison of combined 1/3 km and 5 km shows more complete description, as we expected.*

*As in the response to comment #9, the vertical resolution is 30 m. We then interpolated to 100 m to be consistent with and compared to those from surface observations. These have been added in the revised manuscript.*

11. Page 5, last paragraph of section 2: are the surface products only selected when coincident with an A-train orbit? And, most importantly, are the profiles to be used in section 3 monthly means/accumulations??

*All surface profiles in a month are included in the monthly mean calculation. This is specified in the revised manuscript. "Monthly means are calculated for both surface observations and for the space-based sensors. All heights are above the mean sea level. All surface profiles in a month are accumulated for calculation of monthly means."*

12. Figure 1, 2, 7, 9, 10 and 11: the color bar covers 0-50% but from the text cloud fractions exceed this value at low levels it seems. Why not use the full range of available values?

*Figures 1,2,7,9,10 have been updated in the revised manuscript. The color range extends to 0-80% for Barrow, and 0-60% for Eureka. We also tried extending to 0-100% for both stations, and the details in the figures were not shown as well.*

13. How is the "monthly mean total cloud amount" calculated for each instrument? (e.g. line 28, page 6)

*The following text has been added in the revised manuscript "Total cloud (ice cloud, liquid cloud, mixed phase cloud) amounts are also calculated, as the ratio of number of profiles with cloud (ice cloud, liquid cloud, mixed phase cloud) detected in any layer to the total number of profiles".*

14. Figures 4, 5, 6 need to be redone with either thicker lines or (better) in color, to help distinguish between the different lines. It is really hard to read these as they are.

*Figure 4 and 5 have been updated with lines in color in the revised manuscript. We think the lines in Figure 6 are clear, so we did not update Figure 6.*

15. Page 7, sentence on lines 3-4: this is awkward, since you've already explained that the surface products were described in Shupe (2007, 2011), why not skip this first sentence and add reference to these two studies in the next sentence.

*Revised as the reviewer suggested.*

16. Page 9, line 10: "Major differences" between what? Barrow and Eureka or surface and satellite?

*This paragraph has been revised as the following, "Vertical distributions of ice cloud, liquid cloud, and mixed phase cloud at Eureka from space-based observations show similar patterns above 1 km as those from surface observations (Figure 9). The major differences between surface and space-based observations in the cloud vertical distributions at Eureka (Figure 8d, 8e, 8f, and Figure 9) are similar to those at Barrow (Figure 7, Figure 8a, 8b, and 8c). Major differences between surface and space-based observations include: much less ice and mixed phase cloud in the lowest 1 km from space-based observations; greater liquid cloud, and mixed phase cloud above 2 km in the vertical distributions and annual mean of vertical distributions from space-based observations (Figure not shown); comparable monthly mean total cloud amount, higher ice cloud monthly means, lower liquid cloud monthly means, and higher mixed phase cloud monthly means from surface observations relative to space-based observations. In additions, both satellite and surface observations reveal a key difference to the annual cycles of clouds at Eureka versus Barrow. While both sites have a similar annual cycle of ice cloud occurrence with a relative decrease in summer (Figure 8a, and 8d), there are less frequent liquid-containing clouds at Eureka with the annual maximum of these generally shifted to the autumn. These relative annual cycles explain the key differences in total cloud occurrence fraction over the annual cycle and are explained by generally colder and drier conditions in Eureka relative to Barrow (e.g., Shupe 2011).".*

17. Section 3.2: more information is needed: what is the temporal resolution of the combined product? If monthly means, then this is a combination of the monthly means from surface and satellite? Or are these constructed for coincident observations only? Then how are the two products reconciled in term of surface time average vs satellite spatial average? Line 27: "a complete picture of the " monthly "cloud fraction vertical distribution"?

*The blended product is in monthly means. Line 27 has been revised as the reviewer suggested.*

18. Section 3.3: what is the take-home message for this section?

*The following text has been added in the revised manuscript, "These comparisons indicate that liquid water content monthly means from space-based and surface observations show similar annual evolution with noticeable magnitude differences. The ice water content monthly means from space and surface observations share little similarities in annual evolution or magnitude. Further investigation of these differences is warranted in order to combine these products for a complete vertical distribution of cloud water content".*

19. Conclusions: the first "primary conclusion" is the direct consequence of the known limitations in the CloudSat (surface clutter/low sensitivity) and the CALIPSO (attenuation) instruments. References to other studies should be given. For the second "primary conclusion", I would be inclined to conclude that surface observations perform well, regardless of cloud altitude. For the third conclusion, I would encourage the authors to discuss a bit more the implications for the annual cycle of the satellite based deficiencies. Finally, although I agree that the blended product is more accurate than surface only observations, I think that the real advantage is if one is to calculate heating rates and/or TOA/surface fluxes, this is where this product might make a difference. This should be discussed.

*All the suggestions are well received, and correspondent discussions have been added in the revised manuscript as suggested by the reviewer. In each of the primary conclusions an additional sentence or more has been added to better capture implications and context.*

20. Finally, two papers come to mind to address the very last sentence of the paper, where combined satellite products were used to evaluate cloud impacts in the Arctic in Kay et al (2008) and Kay and Gettelman (2009). The authors might want to mention these results.

*We agree. Kay et al. (2008) and Kay and Gettelman (2009) used combined satellite products. The last sentence of the manuscript suggests that we need combine surface-based and satellite products, in addition to combined satellite products. However, we appreciate the suggestions, and both references have been included in the revised manuscript.*

Typos

1. Abstract, line 24: remove "annual cycle" after "vertical distribution"

2. Line 24, page 3: "negligible surface above 0.96 km" does not make sense, is "clutter" missing?

3. Line 26, page 6: please add "to" before "penetrate" and "thick" after "optically"

4. Line 34, page 6: replace "the" before "CloudSat" with "that".

5. Line 13, page 7: add "with" before "2B-CLDCLASS-lidar"

6. Page 8, line 21: remove "This" after "Whether"

7. Page 8. Lone 23: "the" instead of "he" before "whole Arctic"

Kay, J. E., and A. Gettelman (2009), Cloud influence on and response to seasonal Arctic sea ice loss, J. Geophys. Res., 114, D18204, doi:10.1029/2009JD011773

Kay, J. E., T. L'Ecuyer, A. Gettelman, G. Stephens, and C. O'Dell (2008), The contribution of cloud and radiation anomalies to the 2007 Arctic sea ice extent minimum, Geophys. Res. Lett., 35, L08503, doi:10.1029/2008GL033451

*All the typos have been corrected. Both references have been added in the revised manuscript.*

The response to the reviewers' comments is in italic.

Abhay Devasthale, SMHI, Norrköping, Sweden 2017-02-06

**Review of Liu et al. doi:10.5194/acp-2016-1132, 2017**

While I am a core satellite believer, I do understand and appreciate the importance of in-situ measurements, especially in anchoring space based observations. And there is perhaps no other region in the world where we desperately need more in-situ observations than in the Arctic. Combining these two (space based and in-situ) observing systems is even better. So I really appreciate the work done by the authors in this regard. I have few issues mentioned below that I regard minor in nature, but need to be explained/elaborated. I also had an opportunity to go through the comments posted by the other reviewer and I broadly agree with her/him and I hope the authors will address them as well.

*We appreciate Dr. Devasthale's valuable comments. The manuscript becomes better with revisions in response to reviewer's comments and suggestions. We have responded to other reviewer's comments point by point, and made correspondent revisions in the revised manuscript.*

1) The authors discuss a great deal about how they compute vertical cloud fraction, but very little (or almost nothing if I haven't missed anything obvious) about the spatial (and temporal) collocation of space based and in-situ measurements. The impact of uncertainties arising from these issues is not be underestimated, especially when you compare and combined products with different spatial resolutions (even at monthly mean scale). Let's say that you (or CALIPSO team) use 15 CALIOP single shots (1/3 km each, 5x3) to generate 5 km product. What happens when this 5 km product is not centered over Barrow or Eureka and you are inconsistently selecting single shots? Have the authors evaluated few individual cases manually to check what to expect when they merge 1/3, 1 and 5 km data with reference to the station in question?

*We totally agree with the reviewer's comments, and thank for his insight. These issues, e.g. cloud frequency from surface observations v.s. spatial coverage from space-based observations, different spatial resolutions, viewing angles, vertical resolution among satellite products, all contribute to the shown differences in this manuscript. By using long-term observations, e.g. over 4 years at Eureka and over 2 years at Barrow (all data we have right now), we believe the temporal and spatial average would mitigate these issues. When longer term data from both surface-based and space-based are available, it is worth to revisit this, and see how the differences would change.*

*Inspired by the reviewer's comments, we add a paragraph in the "Conclusion" as the following, "Cloud frequency from surface is calculated in the temporal domain, while the cloud fraction from space-based observations is calculated in the spatial domain although near the surface sites. Differences in spatial resolution, viewing angles, vertical resolution, instrument sensitivity to clouds and retrieval algorithms may all contribute to the differences in the cloud vertical distributions from different instruments. Long-term averages of products may mitigate the impacts of some of these factors. Causes of the remaining differences are worth further investigation.*
*"*

2) It would be helpful if the authors also provide some physical explanation of the seasonal highs and lows in cloud fractions seen in the results. For example, in the case of Barrow, why is cloud

fraction peaking in Feb, Apr and Oct months? Why is there a minimum in Jun and Jul? This is different from Eureka. Why? Perhaps Shupe et al (2011; 2015) already discuss this, but I think the reader still needs at least a brief description of it to make full sense of the differences you observe from these two observing systems.

*A short description of the difference between Barrow and Eureka has been added to the end of Section 3.1.2. This explanation also links to a more detailed discussion of the matter in Shupe (2011). The discussion is "In additions, both satellite and surface observations reveal a key difference to the annual cycles of clouds at Eureka versus Barrow. While both sites have a similar annual cycle of ice cloud occurrence with a relative decrease in summer (Figure 8a, and 8d), there are less frequent liquid-containing clouds at Eureka with the annual maximum of these generally shifted to the autumn. These relative annual cycles explain the key differences in total cloud occurrence fraction over the annual cycle and are explained by generally colder and drier conditions in Eureka relative to Barrow (e.g., Shupe 2011)".*

*These findings have also been added in the abstract and conclusions.*

3) In the case of Barrow station, I am bit surprised at the differences in CF between 2B-GEORPFO and 2B-GEORPOF-Lidar in Aug (Fig. 2). When you add CALIOP there seems to be increase in clouds in the free troposphere from 1 to 5 km. Instinctively, I would have thought that, in the free troposphere, CALIOP would add those subvisual or super thin clouds that are missed by CPR, located in the upper troposphere lower stratosphere. Nearly 30-40% more clouds are added by GEOPROF-Lidar compared to GEOPROF in the lower and middle troposphere and it seems that even surface measurements missed these clouds. Even more confusing is the fact that CALIPSO 5 km doesn't show these clouds in Aug. So what is happening here? Part of this discrepancy can be due to the attenuation of CALIOP signal and part of it due to high amount thin clouds in the middle and lower troposphere (Devasthale et al. 2011). But it is difficult to say without further investigations.

*I agree with the reviewer's comment. The GEOPROF-Lidar has higher values than the sum of those from 2B-GEOPROF and CALIPSO 5 km in August at Barrow. The reviewer gave some possible causes, and we appreciated that and have included such discussion in the revised manuscript. However, it is still unclear why the 2B-GEOPROF-lidar has higher values than the sum of those from 2B-GEOPROF and CALIPSO 5 km. Though finding the causes is beyond the scope of this study, it is worth further investigation in future work. The following discussion has been added in the revised manuscript.*
*"It is worth pointing out that the 2B-GEOPROF-LIDAR shows higher cloud amount values from 1 km to 5 km in the troposphere than the sum of cloud amounts from 2B-GEOPROF and CALIPSO 5 km. The differences can be partially attributed to the attenuation of CALIOP signal and high amount thin clouds in the middle and lower troposphere (Devasthale et al. 2011). Though attribution investigation is beyond the scope of this study, it is worth further investigation in future studies".*

4) The authors say that the blended cloud vertical distribution provides a complete picture. But how do we quantitatively know this? After all, we need a third independent reference to make that conclusion.

*We totally agree. A 3-D cloud distribution product would be ideal with known uncertainties. However, such a product does not exist, and probably will not be available in the near future. So, in my humble opinion, we need to work hard on getting the uncertainties of the existing products, and hopefully merging them for better quality. That is the motivation of this study.*

[revised manuscript text omitted]

* * *
Yinghui Liu 3/31/2017 12:58 PM

Yinghui Liu 3/31/2017 12:58 PM

Matt Shupe 3/29/2017 6:10 PM

leannea 4/24/2017 10:22 PM

Matt Shupe 3/29/2017 6:10 PM

Yinghui Liu 3/31/2017 1:02 PM

leannea 4/24/2017 10:22 PM

leannea 4/24/2017 10:22 PM

This study focuses on further examining and comparing the performance of space-based and surface based radar-lidar observations and retrievals to capture the vertical distribution of cloud properties, including cloud fraction, cloud phase, and cloud water content, at two Arctic atmospheric observatories, Barrow, Alaska and Eureka, Canada. Since cloud phase has been shown to have a particularly strong impact on Arctic cloud radiative effects on the surface (Shupe and Intrieri 2004), it

5 is particularly important to understand how differences in viewing geometry impact observations of different cloud phases. Differences between space-based and surface-based cloud (ice cloud, liquid cloud, and mixed phase cloud) amounts, and cloud ice and liquid water contents are shown in terms of monthly means. Based on the comparison performed here, this study also proposes blended products of cloud property vertical distributions from surface and space-based cloud observations at those two Arctic sites to serve as a best estimate cloud product for model and reanalysis evaluation.

10 Space-based radar and lidar in this paper refer to existing instruments, i.e. the Cloud Profiling Radar (CPR) onboard the *CloudSat* and the Cloud-Aerosol LIdar with Orthogonal Polarization (CALIOP) onboard the Cloud–Aerosol lidar and Infrared Pathfinder Satellite Observation (CALIPSO). However, the conclusions will likely be valid for other space-based radar and lidar instruments, e.g., the ATmospheric backscatter LIDar (ATLID), and the CPR onboard the EarthCARE mission (Hélière et al. 2007).

15 **2 Data and Method**

From the possible Arctic atmospheric observation sites, we have selected Barrow (71°19' N, 156°37' W) and Eureka (80°80' N, 85°57' W) because of the availability of daily cloud vertical profiles from surface observations from 2006 to 2010 when space-based observations are available. The combined radar-lidar cloud fraction best estimation, cloud fraction vertical profiles, cloud phase vertical profiles, and cloud water content vertical profiles, from surface observations at these two sites

20 are described in detail in Shupe et al. (2011), Shupe (2011), and Shupe et al. (2015). These products are based on coincident measurements from the Ka-band cloud radar, depolarization lidars including the micropulse lidar (MPL) at Barrow and the high spectral-resolution lidar (HSRL) at Eureka, microwave radiometer, and radiosondes, which are combined to determine cloud phase (Shupe 2007) and microphysical properties at 1-min temporal and 100-m vertical resolutions.

Observations from *CloudSat* and CALIPSO provide an unprecedented opportunity for a spatially extensive picture of cloud

25 cover in the Arctic (Stephens et al. 2002; Winker et al. 2003). The Vertical Feature Mask (VFM) version 3.01 from CALIPSO's CALIOP provides cloud vertical distribution in up to 10 vertical layers at 5 km and 1 km horizontal resolutions, and up to 5 vertical layers at 1/3 km horizontal resolution (Vaughan et al. 2009). The vertical resolution is 30 m below 8.2 km, and 60 m between 8.2 and 20.2 km. A Selective Iterated BoundarY Location (SIBYL) scheme is applied to detect all features within a given scene. Strongly scattering features, e.g. stratus clouds, can be identified in a single laser pulse, with

30 the 1/3 km horizontal resolution, and these features are then removed in order to detect any surrounding aerosol layers. Weakly scattering features, e.g. thin cirrus clouds, are detected with the average of several laser pulses, e.g. 5 km horizontal resolution, for higher signal-to-noise ratio (Vaughan et al. 2005). Compared to the 1 km resolution data, the 5 km resolution

Yinghui Liu 3/31/2017 1:06 PM

Matt Shupe 3/29/2017 6:11 PM

leannea 4/24/2017 10:23 PM
leannea 4/24/2017 10:23 PM
leannea 4/24/2017 10:23 PM

Yinghui Liu 3/31/2017 1:51 PM

Yinghui Liu 3/22/2017 11:00 AM
Yinghui Liu 3/22/2017 11:00 AM
Yinghui Liu 3/22/2017 11:12 AM
Yinghui Liu 3/22/2017 11:12 AM
Yinghui Liu 3/22/2017 12:19 PM
Matt Shupe 3/29/2017 6:12 PM
Matt Shupe 3/29/2017 6:12 PM
Matt Shupe 3/29/2017 6:12 PM

[revised manuscript text omitted]

Yinghui Liu 3/31/2017 3:46 PM
leannea 4/24/2017 10:26 PM
Yinghui Liu 3/31/2017 3:47 PM
leannea 4/24/2017 10:26 PM
Yinghui Liu 3/23/2017 10:48 AM
Matt Shupe 3/29/2017 6:16 PM
leannea 4/24/2017 10:28 PM
Matt Shupe 3/29/2017 6:17 PM
Yinghui Liu 3/23/2017 10:55 AM
leannea 4/24/2017 10:28 PM
Yinghui Liu 3/31/2017 3:49 PM
leannea 4/24/2017 10:29 PM
Yinghui Liu 3/23/2017 10:58 AM
leannea 4/24/2017 10:29 PM
Yinghui Liu 3/23/2017 10:59 AM
leannea 4/24/2017 10:29 PM
Matt Shupe 3/29/2017 6:17 PM
Matt Shupe 3/29/2017 6:17 PM
leannea 4/24/2017 10:30 PM
Matt Shupe 3/29/2017 6:18 PM
leannea 4/24/2017 10:32 PM
Matt Shupe 3/29/2017 6:19 PM
leannea 4/24/2017 10:32 PM

**3 Results**

**3.1 Cloud fraction vertical distribution**

**3.1.1 Barrow**

[revised manuscript text omitted]

Ieannea 4/25/2017 7:39 AM

Yinghui Liu 4/19/2017 1:41 PM

Ieannea 4/25/2017 7:39 AM

[Figure]

**Figure 9: Vertical Distributions of ice phase cloud**s (1st row), **liquid phase cloud**s (2nd row), **and mixed-phase cloud**s (3rd row) **from 2B-CLDCLASS-lidar (left column), from surface** observations **(middle column), and** the **difference of 2B-CLDCLASS-lidar and surface** observations **at Eureka for 2006-2010.**

[Figure]

Yinghui Liu 3/22/2017 4:17 PM

Yinghui Liu 3/22/2017 4:18 PM

Yinghui Liu 3/22/2017 4:18 PM

Yinghui Liu 3/22/2017 4:18 PM

Yinghui Liu 3/22/2017 4:19 PM

[Figure]

**Figure 10: Blended cloud fraction/frequency vertical distribution at Barrow and Eureka with combined surface and space observations from 2B-GEOPROF-lidar for 2006-2010.**

Yinghui Liu 3/22/2017 4:20 PM

[Figure]

[Figure]

**Figure 11: Blended vertical distributions of (a) ice phase clouds, (b) liquid phase clouds, and (d) mixed-phase clouds at Barrow, and (d) ice phase clouds, (e) liquid phase clouds, and (f) mixed-phase clouds at Eureka from 2B-CLDCLASS-lidar and surface observations for 2006-2010.**

Yinghui Liu 3/22/2017 4:21 PM

Yinghui Liu 3/22/2017 4:22 PM

Yinghui Liu 3/22/2017 4:23 PM

[Figure]

**Figure 12: Vertical Distributions of cloud water content for ice clouds from (a) 2B-CWC-RO, and (b) surface observations, and for liquid clouds from (c) 2B-CWC-RO, and (d) surface observations at Barrow for 2006-2010.**

[Figure]

(a) Ice cloud water content at Eureka (mg m$^{-3}$)

[Figure]

(b) Ice cloud water content, surface, Eureka (mg m$^{-3}$)

[Figure]

(c) Liquid cloud water content at Eureka (mg m$^{-3}$)

[Figure]

(d) Liquid cloud water content, surface, Eureka (mg m$^{-3}$)

**Figure 13: Same as Figure 12, but for Eureka.**